# RanPAC: Random Projections and Pre-trained Models for Continual Learning

**Mark D. McDonnell**[1]**, Dong Gong**[2]**, Amin Parveneh**[1]**,**
**Ehsan Abbasnejad**[1] **and Anton van den Hengel**[1]
[1]Australian Institute for Machine Learning, The University of Adelaide
[2]School of Computer Science and Engineering, University of New South Wales

mark.mcdonnell@adelaide.edu.au, dong.gong@unsw.edu.au,
amin.parvaneh@adelaide.edu.au, ehsan.abbasnejad@adelaide.edu.au,
anton.vandenhengel@adelaide.edu.au

## Abstract

Continual learning (CL) aims to incrementally learn different tasks (such as classification) in a non-stationary data stream without forgetting old ones. Most CL works focus on tackling catastrophic forgetting under a learning-from-scratch paradigm. However, with the increasing prominence of foundation models, pretrained models equipped with informative representations have become available for various downstream requirements. Several CL methods based on pre-trained models have been explored, either utilizing pre-extracted features directly (which makes bridging distribution gaps challenging) or incorporating adaptors (which may be subject to forgetting). In this paper, we propose a concise and effective approach for CL with pre-trained models. Given that forgetting occurs during parameter updating, we contemplate an alternative approach that exploits training-free random projectors and class-prototype accumulation, which thus bypasses the issue. Specifically, we inject a frozen Random Projection layer with nonlinear activation between the pre-trained model's feature representations and output head, which captures interactions between features with expanded dimensionality, providing enhanced linear separability for class-prototype-based CL. We also demonstrate the importance of decorrelating the class-prototypes to reduce the distribution disparity when using pre-trained representations. These techniques prove to be effective and circumvent the problem of forgetting for both class- and domain-incremental continual learning. Compared to previous methods applied to pre-trained ViT-B/16 models, we reduce final error rates by between 20% and 62% on seven class-incremental benchmark datasets, despite not using any rehearsal memory. We conclude that the full potential of pre-trained models for simple, effective, and fast continual learning has not hitherto been fully tapped. Code is available at https://github.com/RanPAC/RanPAC.

## 1 Introduction

Continual Learning (CL) is the subfield of machine learning within which models must learn from a distribution of training samples and/or supervision signals that change over time (often divided into a distinct set of $T$ episodes/tasks/stages) while remaining performant on anything learned previously during training [51, 53]. Traditional training methods do not work well for CL because parameter updates become biased to newer samples, overwriting what was learned previously. Moreover, training on sequential disjoint sets of data means there is no opportunity to learn differences between samples from different stages [48]. These effects are often characterised as 'catastrophic forgetting' [51].

37th Conference on Neural Information Processing Systems (NeurIPS 2023).

Although many methods for CL have been proposed [51, 53, 2, 60], most focus on models that need to be trained from scratch, with resulting performance falling short of that achievable by non-CL alternatives on the same datasets. Although valid use cases for training from scratch will always exist, the new era of large foundation models has led to growing interest in combining CL with the advantages of powerful pre-trained models, namely the assumption that a strong generic feature-extractor can be adapted using fine-tuning to any number of downstream tasks.

Although forgetting still occurs in CL with such transfer-learning [57, 65, 63] (whether using conventional fine-tuning [24] or more-recent Parameter-Efficient Transfer Learning (PETL) methods such as [6, 28, 21]), commencing CL with a powerful feature-extractor has opened up new ideas for avoiding forgetting that are unlikely to work when training from scratch. Such new methods have been applied successfully to pre-trained transformer networks [39, 10, 57, 56, 46, 20, 65, 63, 47, 55], CNNs [11, 15, 33, 22, 59, 37] and multi-modal vision-language models [8].

Three main strategies are evident in these proposals (see Section 2) for details): (i) prompting of transformer networks; (ii) careful selective fine-tuning of a pre-trained model's parameters; and (iii) Class-Prototype (CP) accumulation. Common to all these strategies is the absence of a need for a buffer of rehearsal samples from past tasks, unlike the best CL methods for training models from scratch. Instead, these strategies leverage a pre-trained model's strong feature extraction capabilities.

Each strategy has yielded empirically comparable performance when used with the same benchmarks and pre-trained model (e.g. a ViT B/16 transformer network [46, 63, 65]). It therefore remains open as to what strategy best leverages pre-trained foundation models for CL, in terms of performance on diverse datasets and CL scenarios, simplicity, and efficiency. However, we note that the CP-based CL strategy is simple to apply to both CNNs and transformer networks, whereas prompting methods rely on a prepending learnable prompts to transformer network inputs. Fine-tuning a pre-trained model's parameters requires more resources for training than the other two strategies, while carrying a greater risk of cumulative forgetting over time, thus requiring use of additional CL methodologies.

In this paper, we show that the CP strategy has not come close to exhausting its capacity for accuracy, and can achieve standout performance with carefully tailored strategies to enhance the extracted feature representations from the pre-trained models. CP-based methods use only the prototypes obtained by averaging the extracted features to represent each class, subject to a discrepancy with the data distributions in practice. We propose to handle this via training-free frozen random projections and a decorrelation process, both of which bypass the forgetting issue. Specifically, we introduce a Random-Projection (RP) layer of frozen untrained weights, with nonlinear activation, between the pre-trained-model's feature layer, and a CP-based output head.

**Nonlinear Random Projection (RP) of Features:** The original RP idea (see Figure 1) has been proposed [43, 4, 18] and applied (e.g. [32]) multiple times in the past, as a standalone non-continual learner. Our two-fold motivation for using the same method for CL differs from the method's origins (outlined in Section 2). Firstly, we show that past CP strategies for CL with pre-trained models are equivalent to the linear models learned following RP in the past, e.g. [43, 4, 18, 32]. Secondly, the frozen untrained/training-free weights do not cause forgetting in CL. These two observations suggest using the RP idea alongside CP strategies for CL.

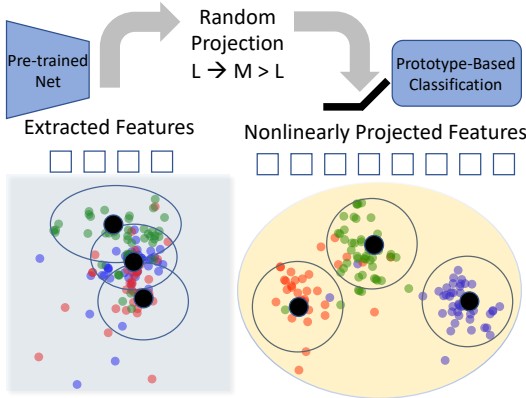

Figure 1: The RP method can lead to a representation space with clear class separation. Colored points are 2D t-SNE visualizations for CIFAR-100 classes with features from a pre-trained ViT-B/16 transformer network.

However, why should RPs provide any value instead of trying to learn a layer of fully-connected weights? It is well known that linear models can benefit from transformations that create nonlinear feature interactions. Our hypothesis in proposing to use RP followed by nonlinear-activation for CL, therefore, is that the transformed set of features is more linearly separable using a CP method than features directly extracted from a pre-trained model. The past work of [43, 4, 18, 32] leads us to expect that for this to be confirmed, RP will need to increase dimensionality from $L$ to $M > L$.

Our **contributions** are summarised as follows:

1. We examine in detail the CP strategy for CL with pre-trained networks, show that it benefits from injecting a Random Projection layer followed by nonlinear activation, and illustrate why. We also analyse why it is important to follow the lead of [37] to linearly transform CPs, via decorrelation using second-order feature statistics.

2. We show that random projections are particularly useful when also using PETL methods with first-session training (see Section 2.1). Accuracy with this combination approaches the CL joint training upper bound on some datasets (Table 2). For ViT-B/16 pre-trained models, we report the highest-to-date rehearsal-free CL accuracies on all class-incremental and domain-incremental datasets we tested on, with large margins when compared to past CP strategies.

3. We highlight the flexibility of our resulting CL algorithm, RanPAC; it works with arbitrary feature vectors (e.g. ViT, ResNet, CLIP), and is applicable to diverse CL scenarios including class-incremental (Section 5), domain-incremental (Section 5) and task-agnostic CL (Appendix F).

## 2 Related Work

### 2.1 Three strategies for CL with strong pre-trained models

**Prompting of transformer networks**: Using a ViT-B/16 network [9], Learning To Prompt (L2P) [57], and DualPrompt [56] reported large improvements over the best CL methods that do not leverage pre-trained models, by training a small pool of prompts that update through the CL process. CODA-Prompt [46], S-Prompt [55] and PromptFusion [5] then built on these, showing improvements in performance.
**Careful fine-tuning of the backbone**: SLCA [63] found superior accuracy to prompt strategies by fine-tuning a ViT backbone with a lower learning rate than in a classifier head. However, it was found that use of softmax necessitated introduction of a 'classifier alignment' method, which incurs a high memory cost, in the form of a feature covariance matrix for every class. Another example of this strategy used selected fine-tuning of some ViT attention blocks [47], combined with traditional CL method, L2 parameter regularization. Fine-tuning was also applied to the CLIP vision-language model, combined with well-established CL method, LwF [8].
**Class-Prototype (CP) accumulation**: Subsequent to L2P, it was pointed out for CL image classifiers [20] that comparable performance can be achieved by appending a nearest class mean (NCM) classifier to a ViT model's feature outputs (see also [39]). This strategy can be significantly boosted by combining with Parameter-Efficient Transfer Learning (PETL) methods (originally proposed for NLP models in a non-CL context [17, 12]) trained only on the first CL stage ('first-session training') to bridge any domain gap [65, 37]. The three PETL methods considered by [65] for transformer networks, and the FiLM method used by [37] for CNNs have in common with the first strategy (prompting) that they require learning of new parameters, but avoid updating any parameters of the backbone pre-trained network. Importantly, [37] also showed that a simple NCM classifier is easily surpassed in accuracy by also accumulating the covariance matrix of embedding features, and learning a linear classifier head based on linear discriminant analysis (LDA) [35]. The simple and computationally lightweight algorithm of [37] enables CL to proceed after the first session in a perfect manner relative to the union of all training episodes, with the possibility of catastrophic forgetting avoided entirely.

CPs are well suited to CL generally [42, 7, 31, 16] and for application to pre-trained models [20, 65, 37], because when the model from which feature vectors are extracted is frozen, CPs accumulated across $T$ tasks will be identical regardless of the ordering of the tasks. Moreover, their memory cost is low compared with using a rehearsal buffer, the strategy integral to many CL methods [53].

### 2.2 RP method for creating feature interactions

As mentioned, the original non-CL usage of a frozen RP layer followed by nonlinear projection as in [43, 4, 18] had different motivations to us, characterized by the following three properties. First, keeping weights frozen removes the computational cost of training them. Second, when combined with a linear output layer, the mean-square-error-optimal output weights can be learned by exact numerical computation using all training data simultaneously (see Appendix B.3) instead of iteratively. Third, nonlinearly activating random projections of randomly projected features is motivated by the

assumption that nonlinear random interactions between features may be more linearly separable than the original features. Analysis of the special case of pair-wise interactions induced by nonlinearity can be found in [32], and mathematical properties for general nonlinearities (with higher order interactions) have also been discussed extensively, e.g. [4, 18].

## 3 Background

### 3.1 Continual learning problem setup

We assume the usual supervised CL setup of a sequence of $T$ tasks/stages, $\mathcal{D} = \{\mathcal{D}_1, \ldots \mathcal{D}_T\}$. In each $\mathcal{D}_t$, a disjoint set of training data paired with their corresponding labels is provided for learning. Subsequent stages cannot access older data. We primarily consider both 'Class-Incremental Learning' (CIL) and 'Domain-Incremental learning' (DIL) protocols [51] for classification of images. In CIL, the class labels for each $\mathcal{D}_t$ are disjoint. One of the challenging aspects of CIL is that, in contrast to Task-Incremental Learning (TIL), the task identity and, consequently, the class subset of each sample is unknown during CIL inference [51]. In DIL, while all stages typically share the same set of classes, there is a distribution shift between samples appearing in each stage. For example, $\mathcal{D}_1$ may include photographs and $\mathcal{D}_2$ images of paintings.

We introduce $K$ as the total number of classes considered within $T$ tasks, and the number of training samples in each task as $N_t$ with $N := \sum_{t=1}^{T} N_t$. For the $n$–th unique training sample within task $\mathcal{D}_t$, we use $\mathbf{y}_{t,n}$ as its length $K$ one-hot encoded label and $\mathbf{f}_{t,n} \in \mathbb{R}^L$ to denote features extracted from a frozen pre-trained model. We denote by $\mathbf{f}_{\text{test}}$ the encoded features for a test instance for which we seek to predict labels.

### 3.2 Class-Prototype strategies for CL with pre-trained models

For CL, using conventional cross-entropy loss by linear probing or fine-tuning the feature representations of a frozen pre-trained model creates risks of task-recency bias [31] and catastrophic forgetting. Benefiting from the high-quality representations of a pre-trained model, the most straightforward Class-Prototype (CP) strategy is to use Nearest Class Mean (NCM) classifiers [58, 34], as applied and investigated by [65, 20]. CPs for each class are usually constructed by averaging the extracted feature vectors over training samples within classes, which we denote for class $y$ as $\bar{\mathbf{c}}_y$. In inference, the class of a test sample is determined by finding the highest similarity between its representation and the set of CPs. For example, [65] use cosine similarity to find the predicted class for a test sample,

$$y_{\text{test}} = \arg \max_{y' \in \{1,\ldots,K\}} s_{y'}, \quad s_y := \frac{\mathbf{f}_{\text{test}}^\top \bar{\mathbf{c}}_y}{||\mathbf{f}_{\text{test}}|| \cdot ||\bar{\mathbf{c}}_y||}. \tag{1}$$

However, it is also not difficult to go beyond NCM within the same general CL strategy, by leveraging second-order feature statistics [11, 37]. For example, [37] finds consistently better CL results with pre-trained CNNs than NCM using an incremental version [36, 11] of Linear Discriminant Analysis (LDA) classification [35], in which the covariance matrix of the extracted features is continually updated. Under mild assumptions, LDA is equivalent to comparing feature vectors and class-prototypes using Mahalanobis distance (see Appendix B.4), i.e. different to cosine distance used by [65].

### 3.3 CP-based classification using Gram matrix inversion

We will also use incrementally calculated second-order feature statistics, but create a simplification compared with LDA (see Appendix B.4), by using the Gram matrix of the features, $\mathbf{G}$, and $\mathbf{c}_y$ (CPs with averaging dropped), to obtain the predicted label

$$y_{\text{test}} = \arg \max_{y' \in \{1,\ldots,K\}} s_{y'}, \quad s_y := \mathbf{f}_{\text{test}}^\top \mathbf{G}^{-1} \mathbf{c}_y. \tag{2}$$

Like LDA, but unlike cosine similarity, this form makes use of a training set to 'calibrate' similarities. This objective has a basis in long established theory for least square error predictions of one-hot encoded class labels [35] (see Appendix B.3). Similar to incremental LDA [36, 11], during CL training, we describe in Section 4.3 how the Gram matrix and the CPs corresponding to $\mathbf{c}_y$ can easily be updated progressively with each task. Note that Eqn. (2) is expressed in terms of the maximum number of classes after $T$ tasks, $K$. However, for CIL, it can be calculated after completion of tasks $t < T$ with fewer classes than $K$. For DIL, all $K$ classes are often available in all tasks.

# 4 The proposed approach and theoretical insights: RanPAC

## 4.1 Why second order statistics matter for CPs

We show in Section 5 that Eqn. (2) leads to better results than NCM. We attribute this to the fact that raw CPs are often highly correlated between classes, resulting in poorly calibrated cosine similarities, whereas the use of LDA or Eqn (2) mostly removes correlations between CPs, creating better separability between classes. To illustrate these insights, we use the example of a ViT-B/16 transformer model [9] pre-trained on ImageNet-21K with its classifier head removed, and data from the well-established 200-class Split Imagenet-R CIL benchmark [56].

For comparison with CP approaches, we jointly trained, on all 200 classes, a linear probe softmax classifier. We treat the weights of the joint probe as class-prototypes for this exercise and then find the Pearson correlation coefficients between each pair of prototypes as shown for the first 10 classes in Fig. 2 (right). Compared with the linear probe, very high off-diagonal correlations are clearly observed when using NCM, with a mean value more than twice that of the original ImageNet-1K training data treated in the same way. This illustrates the extent of the domain shift for the downstream dataset. However, these correlations are mostly removed when using Eqn (2). Fig. 2 (left) shows that high correlation coefficients coincide with poorly calibrated cosine similarities between class-prototypes and training samples, both for true class comparisons (similarities between a sample's feature vector and the CP for the class label corresponding to that sample.) and inter-class comparisons (similarities between a sample's feature vector and the class prototypes for the set of N-1 classes not equal to the sample's class label). However, when using Eqn. (2), the result (third row) is to increase the training-set accuracy from 64% to 75%, coinciding with reduced overlap between inter- and true-class similarity distributions, and significantly reduced off-diagonal correlation coefficients between CPs. The net result is linear classification weights that are much closer to those produced by the jointly-trained linear probe. These results are consistent with known mathematical relationships between Eqn (2) and decorrelation, which we outline in Appendix B.4.4.

## 4.2 Detailed overview and intuition for Random Projections

CP methods that use raw $\bar{c}_y$ for CL assume a Gaussian distribution with isotropic covariance. Empirically, we have found that when used with pre-trained models this assumption is invalid (Figure 2). One can learn a non-linear function (e.g. using SGD training of a neural network) with

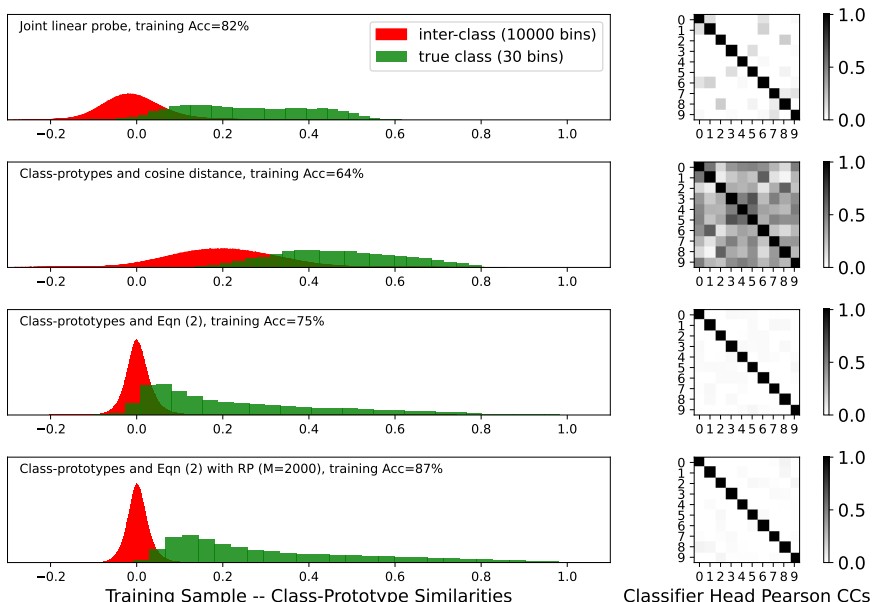

Figure 2: Left: Histograms of similarities between class-prototypes and feature vectors from the ViT-B/16 transformer model pre-trained on ImageNet-1K, for the training samples of the Imagenet-R dataset. Right: Pearson correlation coefficients (CCs) for 10 pairs of class-prototypes. Reduced correlations between CPs of different classes (right), coincides with better class separability (left).

high capacity and non-convex objective to alleviate this gap (e.g., learning to adapt on the first task as in [65, 37]), but that requires additional assumptions and architectural choices. In addition for CL, any learning on all tasks carries a high risk of catastrophic forgetting. Instead, we take advantage of the observation that the objectives in CP strategies principally require firstly feature projections (that could be possibly adjusted for their distribution by a covariance matrix) and secondary the feature prototypes. For the latter, since using large pre-trained models produce features that capture most prominent patterns and attributes, the average is a reasonable expected class representative. However, the feature representations may also benefit from nonlinear transformation to enhance linear separability. In this paper, we consider two points: (i) we show a random projection of the features in the embedding space (or their projection to a random basis) to a higher dimensional space leads to a distribution that is more likely to be suitable for a Gaussian fit (e.g. using Eqn. (2)); and, (ii) when the distribution of representations is nearly Gaussian, we can simply perform a linear regression (or LDA or a conditional mean of the joint Gaussian distribution).

This assertion is supported by the bottom row of Fig. 2, which shows that the application of an RP layer of size $M = 2000$ (chosen arbitrarily, for the purpose of illustration) reduces the overlap between the similarity histograms further than otherwise, and shifts the in-class histogram to the right, coinciding with higher accuracy, surpassing that of the joint linear probe, consistent with our findings in Section 5. We note that these RPs can be generated once and *then frozen for use in all continual learning stages*, enabling a simple and efficient algorithm.

We first analyse the inner products of the features obtained from the pre-trained model projected using some random basis vectors (i.e. random projections). We consider $M$ such random vectors all randomly drawn i.i.d from a Gaussian distribution with variance $\sigma^2$. The number $M$ defines the dimension of the space the random projections of these features constitute. Considering random projections $\mathbf{W} \in \mathbb{R}^{L \times M}$, for any two given feature vectors $\mathbf{f}, \mathbf{f}' \in \mathbb{R}^L$ we have (see Appendix B.2),

$$\mathbb{E}_{\mathbf{W}}\left[(\mathbf{W}^\top \mathbf{f})^\top (\mathbf{W}^\top \mathbf{f}')\right] = \sum_i^M \mathbb{E}_{\mathbf{W}}\left[\mathbf{W}_{(i)}^2\right] \mathbf{f}^\top \mathbf{f}' + \sum_{i \neq j}^M \mathbb{E}_{\mathbf{W}}\left[\mathbf{W}_{(i)}\right]^\top \mathbb{E}_{\mathbf{W}}\left[\mathbf{W}_{(j)}\right] \mathbf{f}^\top \mathbf{f}', \quad (3)$$

where $\mathbf{W}_{(i)}$ denotes the $i$th column. That is, the expected inner products of projections of these two features can be decomposed. Now, we have $\mathbf{W}_{(i)} \sim \mathcal{N}(\mathbf{0}, \sigma^2 \mathbf{I})$, thus $\mathbb{E}_{\mathbf{W}}[\mathbf{W}_{(i)}] = \mathbb{E}_{\mathbf{W}}[\mathbf{W}_{(j)}] = 0$ and the second term in Eqn. (3) vanishes. Further, $\sum_i^M \mathbb{E}_{\mathbf{W}}[\mathbf{W}_{(i)}^2] = M\sigma^2$. We can make two observations (theoretical details are provided in Appendix B.2):

1. As $M$ increases, the likelihood that the norm of any projected feature approaching the variance increases. In other words, the projected vectors in higher dimensions almost surely reside on the boundary of the distribution with almost equal distance to the mean (the distribution approaches isotropic Gaussian).

2. As $M$ increases, it is more likely for angles between two randomly projected instances to be distinct (i.e. the inner products in the projected space are more likely to be larger than some constant).

This discussion can readily be extended to incorporate nonlinearity. The benefit of incorporating nonlinearity is to (i) incorporate interaction terms [54, 50], and, (ii) simulate higher dimensional projections that otherwise could be prohibitively large in number. To see the later point, denoting by $\phi$ the nonlinearity of interest, we have $\phi(\mathbf{W}^\top \mathbf{f}) \approx \hat{\mathbf{W}}^\top \hat{\mathbf{f}}$ where $\hat{\mathbf{f}}$ is obtained from a linear expansion using Taylor series and $\hat{\mathbf{W}}$ is the corresponding projections. The Taylor expansion of the nonlinear function $\phi$ gives rise to higher order interactions between dimensions. Although vectors of interaction terms can be formed directly, as in the methods of [38, 54], this is computationally prohibitive for non-trivial $L$. Hence, the use of nonlinear projections of the form $\mathbf{h}_{\text{test}} := \phi(\mathbf{f}_{\text{test}}^\top \mathbf{W})$ is a convenient alternative, as known to work effectively in a non-CL context [43, 4, 18, 32] .

### 4.3 Random projection for continual learning

Using the random projection discussed above, with $\phi(\cdot)$ as an element-wise nonlinear activation function, given feature sample $\mathbf{f}_{t,n}$ we obtain length $M$ representations for CL training in each task, $\mathbf{h}_{t,n} := \phi(\mathbf{f}_{t,n}^\top \mathbf{W})$ (Fig. 1). For inference, $\mathbf{h}_{\text{test}} := \phi(\mathbf{f}_{\text{test}}^\top \mathbf{W})$ is used in $s_y$ in Eqn. (2) instead of $\mathbf{f}_{\text{test}}$. We define $\mathbf{H}$ as an $M \times N$ matrix in which columns are formed from all $\mathbf{h}_{k,n}$ and for convenience refer to only the final $\mathbf{H}$ after all $N$ samples are used. We now have an $M \times M$ Gram

matrix for the features, $\mathbf{G} = \mathbf{H}\mathbf{H}^\top$. The random projections discussed above are sampled once and left frozen throughout all CL stages.

Like the covariance matrix updates in streaming LDA applied to CL [11, 37], variables are updated either for individual samples, or one entire CL stage, $\mathcal{D}_t$, at a time. We introduce matrix $\mathbf{C}$ to denote the concatenated column vectors of all the $\mathbf{c}_y$. Rather than covariance, $\mathbf{S}$, we update the Gram matrix, and the CPs in $\mathbf{C}$ (the concatenation of $\mathbf{c}_y$'s) iteratively. This can be achieved either one task at a time, or one sample at a time, since we can express $\mathbf{G}$ and $\mathbf{C}$ as summations over outer products as

$$\mathbf{G} = \sum_{t=1}^{T} \sum_{n=1}^{N_t} \mathbf{h}_{t,n} \otimes \mathbf{h}_{t,n}, \qquad \mathbf{C} = \sum_{t=1}^{T} \sum_{n=1}^{N_t} \mathbf{h}_{t,n} \otimes \mathbf{y}_{t,n}. \qquad (4)$$

Both $\mathbf{C}$ and $\mathbf{G}$ will be invariant to the sequence in which the *entirety* of $N$ training samples are presented, a property ideal for CL algorithms.

The mentioned origins of Eqn. (2) in least squares theory is of practical use; we find it works best to use ridge regression [35], and calculate the $l_2$ regularized inverse, $(\mathbf{G} + \lambda\mathbf{I})^{-1}$, where $\mathbf{I}$ denotes the identity matrix. This is achieved methodically using cross-validation—see Appendix C. The revised score for CL can be rewritten as

$$s_y = \phi(\mathbf{f}_{\text{test}}^\top \mathbf{W})(\mathbf{G} + \lambda\mathbf{I})^{-1}\mathbf{c}_y. \qquad (5)$$

In matrix form the scores can be expressed as predictions for each class label as $\mathbf{y}_{\text{pred}} = \mathbf{h}_{\text{test}}\mathbf{W}_\text{o}$. Different to streaming LDA, our approach has the benefits of (i) removing the need for bias calculations from the score; (ii) updating the Gram matrix instead of the covariance avoids outer products of means; and (iii) the form $\mathbf{W}_\text{o} = (\mathbf{G} + \lambda\mathbf{I})^{-1}\mathbf{C}$ arises as a closed form solution for mean-square-error loss with $l_2$-regularization (see Appendix B.3), in contrast to NCM, where no such theoretical result exists. Phase 2 in **Algorithm 1** summarises the above CL calculations.

Application of Eqn. (5) is superficially similar to AdaptMLP modules [6], but instead of a bottleneck layer, we expand to dimensionality $M > L$, since past applications found this to be necessary to compensate for $\mathbf{W}$ being random rather than learned [43, 4, 18, 32]. As discussed in the introduction, the use of a random and training-free weights layer is particularly well suited to CL.

The value of transforming the original features to nonlinear random projections is illustrated in Fig. 3. Features for the $T = 10$ split ImageNet-R CIL dataset were extracted from a pre-trained ViT-B/16 [9] network and Eqn. (5) applied after each of the $T = 10$ tasks. Fig. 3(a) shows the typical CL average accuracy trend, whereby accuracy falls as more tasks are added. When a nonlinear activation, $\phi(\cdot)$ is used (e.g. ReLU or squaring), performance improves as $M$ increases, but when the nonlinear activation is omitted, accuracy is no better than not using RP, even with $M$ very large. On the other hand, if dimensionality is reduced without nonlinearity (in this case from 768 to 500), then performance drops below the No RP case, highlighting that if RPs in this application create dimensionality reduction, it leads to poor performance.

Fig. 3(b) casts light on why nonlinearity is important. We use only the first 100 extracted features per sample and compare application of Eqn. (2) to raw feature vectors (black) and to pair-wise interaction terms, formed from the flattened cross-product of each extracted feature vector (blue trace). Use of the former significantly outperforms the latter. However, when Eqn. (5) is used instead (red trace), the drop in performance compared with flattened cross products is relatively small. Although this suggests exhaustively creating products of features instead of RPs, this is computationally infeasible. As an alternative, RP is a convenient and computationally cheap means to create nonlinear feature interactions that enhance linear separability, with particular value in CL with pre-trained models.

## 4.4 Combining with parameter-efficient transfer learning and first-session adaptation

Use of an RP layer has the benefit of being model agnostic, e.g. it can be applied to any feature extractor. As we show, it can be applied orthogonally to PETL methods. PETL is very appealing for CL, particularly approaches that do not alter any learned parameters of the original pre-trained model, such as [10, 28]. We combine RP with a PETL method trained using CL-compatible 'first-session' training, as carried out by [65, 37]. This means training PETL parameters only on the first CL task, $\mathcal{D}_1$, and then freezing them thereafter (see Phase 1 of **Algorithm 1**). The rationale is that the training data and labels in the first task may be more representative of the downstream dataset than that used to train the original pre-trained model. If a new dataset drifts considerably, e.g. as in DIL, the

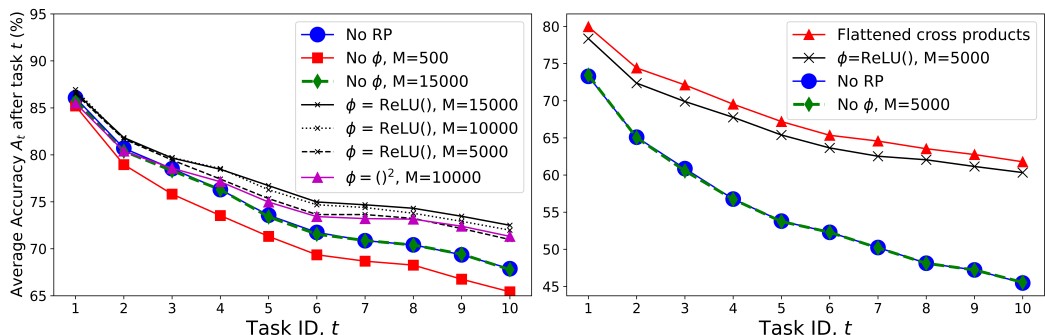

Figure 3: **Impact of RP compared to alternatives.** (a) Using only Phase 2 of **Algorithm 1**, we show average accuracy (see Results) after each of $T = 10$ tasks in the split ImageNet-R dataset. The data illustrates the value of nonlinearity combined with large numbers of RPs, $M$. (b) Illustration that interaction terms created from feature vectors extracted from frozen pre-trained models contain important information that can be mostly recovered when RP and nonlinearity are used.

benefits of PETL may be reduced because of this choice. But the assumption is that the domain gap between the pre-training dataset and the new dataset is significant enough to still provide a benefit. First-session training of PETL parameters requires a temporary linear output layer, learned using SGD with cross-entropy loss and softmax and only $K_1$ classes, which is discarded prior to Phase 2.

For transformer nets, we experiment with the same three methods as [65], i.e. AdaptFormer [6], SSF [28], and VPT [21]. For details of these methods see the cited references and Appendix D. Unlike [65] we do not concatenate adapted and unadapted features, as we found this added minimal value when using RP.

### 4.5 Memory usage of RP layer

The RP layer increases the total memory required by the class prototypes by a factor of $1 + (M - L)/L$, while an additional $LM$ parameters are frozen and untrainable. For typical values of $L = 768$ and $K = 200$, the injection of an RP layer of size $M = 10000$ therefore creates $\sim 10$ times the number of trainable parameters, and adds $\sim 10$M non-trainable parameters. Although this is a significant number of new parameters, it is still small compared with the overall size of the ViT-B/16 model, with its 84 millions parameters. Moreover, the weights of $\mathbf{W}$ can be bipolar instead of Gaussian and if stored using a single bit for each element contribute only a tiny fraction to additional model memory. During training only, we also require updating an $M \times M$ Gram matrix, which is smaller than the $K$ $(L \times L)$ covariance matrices of the SLCA fine-tuning approach [63]

---

**Algorithm 1** RanPAC Training

**Input:** Sequence of $T$ tasks, $\mathcal{D} = \{\mathcal{D}_1, \dots \mathcal{D}_T\}$, pre-trained model, and PETL method
**Phase 1:** 'First-session' PETL adaptation.
    **for** sample $n = 1, \dots, N_1$ in $\mathcal{D}_1$ **do**
        Extract feature vector, $\mathbf{f}_{1,n}$
        Use in SGD training of PETL parameters
    **end for**
**Phase 2:** Continual Learning with RPs.
    Create frozen $L \times M$ RP weights, $\mathbf{W} \sim \mathcal{N}(0, 1)$
    **for** task $t = 1, \dots, T$ **do**
        **for** sample $n = 1, \dots, N_t$ in $\mathcal{D}_t$ **do**
            Extract feature vector, $\mathbf{f}_{t,n}$
            Apply $\mathbf{h}_{t,n} = \phi(\mathbf{f}_{t,n}^\top \mathbf{W})$
            Update $\mathbf{G}$ and $\mathbf{C}$ matrices using Eqn. (4)
        **end for**
        Optimize $\lambda$ (A. C) and compute $(\mathbf{G} + \lambda\mathbf{I})^{-1}$
    **end for**

---

if $M < \sqrt{K}L$. L2P [57], DualPrompt [56] and ADaM [65] each use $\sim$0.3M–0.5M trainable parameters. For $K = 200$ classes and $M = 10000$, RanPAC uses a lot more ($\sim$2–2.5M, depending on the PETL method), comparable to CODA-Prompt [46]. RanPAC also uses 10M untrained parameters, but we highlight that these are not trainable. Moreover, $M$ need not be as high as 10000 (Table A5).

## 5 Experiments

We have applied **Algorithm 1** to both CIL and DIL benchmarks. For the pre-trained model, we experiment mainly with two ViT-B/16 models [9] as per [65]: one self-supervised on ImageNet-21K,

and another with supervised fine-tuning on ImageNet-1K. Comparisons are made using a standard CL metric, Average Accuracy [30], which is defined as $A_t = \frac{1}{t}\sum_{i=1}^{t} R_{t,i}$, where $R_{t,i}$ are classification accuracies on the $i$–th task, following training on the $t$-th task. We report final accuracy, $A_T$, in the main paper, with analysis of each $A_t$ and $R_{t,i}$ left for Appendix F, along with forgetting metrics and analysis of variability across seeds. Appendix F also shows that our method works well with ResNet and CLIP backbones.

We use split datasets previously used for CIL or DIL (see citations in Tables 1 and 3); details are provided in Appendix E. We use $M = 10000$ in **Algorithm 1** except where stated; investigation of scaling with $M$ is in Appendix F.5. All listed 'Joint' results are non-CL training of comparisons on the entirety of $\mathcal{D}$, using cross-entropy loss and softmax.

Key indicative results for CIL are shown in Table 1, for $T = 10$, with equally sized stages (except VTAB which is $T = 5$, identical to [65]). For $T = 5$ and $T = 20$, see Appendix F. For each dataset, our best method surpasses the accuracy of prompting methods and the CP methods of [20] and [65], by large margins. Ablations of **Algorithm 1** listed in Table 1 show that inclusion of our RP layer results in error-rate reductions of between 11% and 28% when PETL is used. The gain is reduced otherwise, but is $\geq 8\%$, except for VTAB. Table 1 also highlights the limitations of NCM.

| Method | CIFAR100 | IN-R | IN-A | CUB | OB | VTAB | Cars |
|---|---|---|---|---|---|---|---|
| Joint linear probe | 87.9% | 72.0% | 56.6% | 88.7% | 78.5% | 86.7% | 51.7% |
| L2P [57] | 84.6% | 72.4% | 42.5% | 65.2% | 64.7% | 77.1% | 38.2%* |
| DualPrompt [56] | 84.1% | 71.0% | 45.4% | 68.5% | 65.5% | 81.2% | 40.1%* |
| CODA-Prompt [46] | 86.3% | 75.5% | 44.5% | 79.5% | 68.7% | 87.4% | 43.2% |
| ADaM [65] | 87.6% | 72.3% | 52.6% | 87.1% | 74.3% | 84.3% | 41.4** |
| Ours (**Algorithm 1**) | **92.2%** | **77.9%** | **62.4%** | **90.3%** | **79.9%** | **92.2%** | **77.5%** |
| Rel. ER vs ADaM$^\dagger$ | ↓ 36% | ↓ 20% | ↓ 21% | ↓ 25% | ↓ 22% | ↓ 53% | ↓ 62% |
| Ablations | | | | | | | |
| No RPs | 90.6% | 73.9% | 57.7% | 86.6% | 74.3% | 90.0% | 73.0% |
| No Phase 1 | 89.0% | 71.8% | 58.2% | 88.7% | 78.2% | 92.2% | 66.0% |
| No RPs or Phase 1 | 87.0% | 67.6% | 54.4% | 86.8% | 73.1% | 91.9% | 62.2% |
| NCM with Phase 1 | 87.8% | 70.1% | 49.7% | 85.4% | 73.4% | 88.2% | 40.5% |
| NCM only | 83.4% | 61.2% | 49.3% | 85.1% | 73.1% | 88.4% | 37.7% |
| Rel. ER (inc. Phase 1) | ↓ 17% | ↓ 15% | ↓ 11% | ↓ 28% | ↓ 22% | ↓ 22% | ↓ 17% |
| Rel. ER (no Phase 1) | ↓ 15% | ↓ 13% | ↓ 8% | ↓ 14% | ↓ 19% | ↓ 4% | ↓ 10% |

Table 1: **Comparison of prompting and CP strategies for CIL:** 'Rel. ER' ($^\dagger$) is Relative Error Rate. No rehearsal buffer is used for any method listed. IN-R is ImageNet-R [56]. The versions of ImageNet-A (IN-A), OmniBenchmark (OB), CUB and VTAB are those defined by [65]. L2P and DualPrompt results are those stated within [65], except * (Cars) from [63]. CODA-Prompt is directly from [46] for CIFAR100 and IN-R, and from our own experiments with the PILOT toolbox [49] otherwise. For ADaM, we have used the best reported by [65] across all PETL methods, except for ** (Cars) which is our own. Remaining results are our own implementation. PromptFusion [5] is omitted due to requiring a rehearsal buffer. Using a NCM method, [20] achieved 83.7% for CIFAR100 and 64.3% for Imagenet-R. **Ablations of Algorithm 1:** NCM uses cosine similarity instead of Eqn. (2); RP means Random Projections layer; Phase 1 is defined within **Algorithm 1**.

**Algorithm 1** surpasses the jointly trained linear probe, on all seven datasets; Table 1 indicates that for all datasets this can be achieved by at least one method that adds additional weights to the pre-trained model. However, NCM alone cannot match the joint linear probe. As shown in Table 2, **Algorithm 1** also does better than fine-tuning strategies for CIL [47, 63]. This is tabulated separately from Table 1 because fine-tuning has major downsides as mentioned in the Introduction. Table 2 also shows Algorithm 1 reaches within 2% raw accuracy of the best joint fine-tuning accuracy on three datasets, with a larger gap on the two ImageNet variants and Cars.

Results for DIL datasets, and corresponding ablations of Algorithm 1 are listed in Table 3. CORe50 is $T = 8$ stages with 50 classes [29], CDDB-Hard is $T = 5$ with 2 classes [27] and DomainNet is $T = 6$ with 345 classes [40] (further details are provided in Appendix E). The same trends as for CIL can be observed, i.e. better performance than prompting strategies and highlighting of the value of the RP layer. For DomainNet, the value of RP is particularly strong, while including PETL adds little value, consistent with the nature of DIL, in which different CL stages originate in different domains.

| Method | CIFAR100 | IN-R | IN-A | CUB | OB | VTAB | Cars |
|---|---|---|---|---|---|---|---|
| Joint full fine-tuning | 93.8% | 86.6% | 70.8% | 90.5% | 83.8% | 92.0% | 86.9% |
| SLCA [63] | 91.5% | 77.0% | - | 84.7%* | - | - | 67.7% |
| Ours (**Algorithm 1**) | **92.2**% | **78.1**% | **61.8**% | **90.3**% | **79.9**% | **92.6**% | **77.7**% |

Table 2: **Comparison with fine-tuning strategies for CIL.** Results for SLCA are directly from [63]. 'Joint' means full fine-tuning of the entire pre-trained ViT network, i.e. a continual learning upper bound. Notes: L2 [47] achieved 76.1% on Imagenet-R, but reported no other ViT results. Cells containing - indicates SLCA did not report results, and no codebase is available yet. Superscript *: the number of CUB training samples used by [63] is smaller than that defined by [65], which we use.

| Method | CORe50 | CDDB-Hard | DomainNet |
|---|---|---|---|
| L2P [37] | 78.3% | 61.3% | 40.2% |
| S-iPrompts [55] | 83.1% | 74.5% | 50.6% |
| ADaM | 92.0% | 70.7% | 50.3% |
| Ours (**Algorithm 1**) | **96.7**% | **86.2**% ($M = 5K$) | **66.6**% |
| S-liPrompts [55] (not ViT B/16) | *89.1%* | *88.7%* | *67.8%* |
| Ablations | | | |
| No RPs | 94.8% | 84.2% | 55.2% |
| No Phase 1 | 94.3% | 81.6% ($M = 5K$) | 64.7% |
| No RPs or Phase 1 | 92.4% | 80.7% | 54.1% |
| NCM with Phase 1 | 91.4% | 73.2% | 51.3% |
| NCM only | 80.4% | 69.8% | 46.4% |
| Rel. ER (inc. Phase 1) | ↓ 37% | ↓ 13% | ↓ 25% |
| Rel. ER (no Phase 1) | ↓ 25% | ↓ 5% | ↓ 23% |

Table 3: **Comparison with prompting and CP strategies for DIL:** No rehearsal buffer is used for any method. The first four rows use ViT B/16 networks. S-liPrompts uses vision-language prompting with a pre-trained CLIP model, and cannot be directly compared to ours. For CORe50, 83.2% was reported for simple CP strategy of [20] and 85.4% for FSA-FiLM with pre-trained convnets [37]. For DomainNet we and [55] use 365 classes, different to [46]. ADaM results are based on code from [65]. L2P and S-prompts are taken from [55]. **Ablations:** Explanations are the same as in Table 1.

# 6   Conclusion

We have demonstrated that feature representations extracted from pre-trained foundation models such as ViT-B/16 have not previously achieved their full potential for continual learning. Application of our simple and rehearsal-free class-prototype strategy, RanPAC, results in significantly reduced error rates on diverse CL benchmark datasets, without risk of forgetting in the pre-trained model. These findings highlight the benefits of CP strategies for CL with pre-trained models.

**Limitations:** The value of Eqs (4) and (5) are completely reliant on supply of a good generic feature extractor. For this reason, they are unlikely to be as powerful if used in CL methods that train networks from scratch. However, it is possible that existing CL methods that utilise self-supervised learning, or otherwise create good feature extractor backbones could leverage similar approaches to ours for downstream CL tasks. As discussed in Section 4.5, RanPAC uses additional parameters compared to methods like L2P. However, this is arguably worth trading-off for the simplicity of implementation and low-cost training.

**Future Work:** Examples in Appendix F shows that our method works well with other CL protocols including: (i) task-agnostic, i.e. CL without task boundaries during training (e.g. Gaussian scheduled CIFAR-100), (ii) use of a non one-hot-encoded target, e.g. language embeddings in the CLIP model, and (iii) regression targets, which requires extending the conceptualisation of class-prototypes to generic feature prototypes. Each of these has a lot of potential extensions and room for exploration. Other interesting experiments that we leave for future work include investigation of combining our approach with prompting methods, and (particularly for DIL) investigation of whether training PETL parameters beyond the first session is feasible without catastrophic forgetting. Few-shot learning with pre-trained models [1, 45] may potentially also benefit from a RanPAC-like algorithm.

## Acknowledgements

This work was supported by the Centre for Augmented Reasoning at the Australian Institute for Machine Learning, established by a grant from the Australian Government Department of Education. Dong Gong is the recipient of an Australian Research Council Discovery Early Career Award (project number DE230101591) funded by the Australian Government. We would like to thank Sebastien Wong of DST Group, Australia, and Lingqiao Liu of the Australian Institute for Machine Learning, The University of Adelaide, for valuable suggestions and discussion.

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

# Appendices

## Appendix A  Overview of RanPAC and comparison to other strategies

Figure A1 provides a graphical overview of the two phases in **Algorithm 1**. Table A1 provides a summary of different strategies for leveraging pre-trained models for Contial Learning (CL) and how our own method, RanPAC, compares.

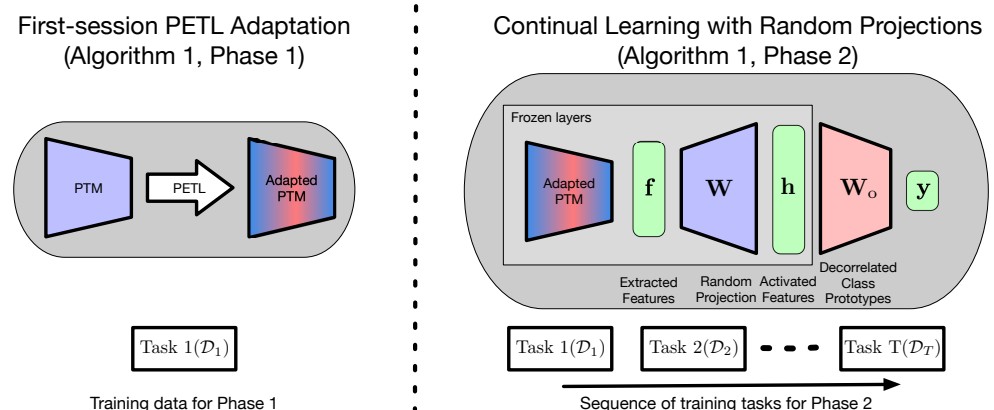

Figure A1: **Overview of RanPAC for CL classification.** In Phase 1 of **Algorithm 1** we optionally inject parameters for a Parameter-Efficient Transfer Learning (PETL) method into a frozen pre-trained model (PTM). The PETL parameters are trained only on Task 1 (the 'first-session') of a set of $T$ continual learning tasks, to help bridge the domain gap, as in [65, 37]. Then in Phase 2, first, $L$-dimensional feature vectors, $\mathbf{f}$, are extracted from the network after completion of learning in Phase 1 (now frozen). Then, the extracted feature vectors are randomly projected to dimension $M$ (typically $M > L$) using frozen weights $\mathbf{W}$, followed by nonlinear activation, $\phi$, to obtain new feature vectors, $\mathbf{h} = \phi(\mathbf{f}^\top \mathbf{W})$. Training in Phase 2 is comprised of continual iterative updating of class prototypes and the Gram matrix, followed by a matrix inversion to compute $M \times N$ matrix $\mathbf{W}_\mathrm{o}$, following the end of each Task's training. These weights can be thought of as decorrelated class prototypes that linearly weight the features in $\mathbf{h}$ to obtain class predictions in $\mathbf{y}$.

| | **Prompting** [57, 56, 46, 55] | **Fine-tuning** [63, 47] | **CP** [20, 65, 37] | **CP + RP** RanPAC |
|---|:---:|:---:|:---:|:---:|
| No Rehearsal Buffer | ✓ | ✓ | ✓ | ✓ |
| Pre-trained model frozen | ✓ | | ✓ | ✓ |
| Transformers and CNNs | | ? | ✓ | ✓ |
| Simplicity | | | ✓ | ✓ |
| Parameter-Efficient | ✓ | | ✓ | ✓ |
| Theoretical support | | | | ✓ |
| SOTA Performance | | | | ✓ |

Table A1: **Comparison of different strategies.** In Section 1, we categorised existing methods for leveraging pre-trained models for CL into three strategies: Prompting, Fine-tuning and Class-Prototypes (CP). Our own method, RanPAC, is a CP strategy in which we introduce a frozen Random Projection (RP) layer with nonlinear activation. Overall, we argue that CP methods provide the best combination of benefits, and within existing CP methods, RanPAC provides the strongest results, complemented by theoretical support for its use of RP and second-order statistics to decorrelate CPs. We note that rehearsal buffers have been used with some of the cited methods to boost performance, and could potentially also do so for methods where they have not yet been used, such as RanPAC.

## Appendix B  Theoretical support

### B.1  Chernoff bound of the norm of the projected vectors

We provide further details for the discussion in Section 4.2. The norm of the vector projected using the Chernoff Bound can be written as:

$$\mathbb{P}\left(\left|\,\|\mathbf{W}^\top\mathbf{f}\| - \mathbb{E}_\mathbf{W}\left[\|\mathbf{W}^\top\mathbf{f}\|\right]\,\right| > \epsilon\sigma^2\right) \le 2\exp\left(-\frac{\epsilon^2\sigma^2}{2M+\epsilon}\right). \tag{6}$$

This bound indicates the relation between the dimensionality and the expected variation in the norm of the projected vectors. For fixed $\sigma$ and $\epsilon$, as $M$ increases, the right-hand side approaches 1, indicating that it is more likely for the norm of the projected vector to be in a desired distance to the expectation. In other words, these projected vectors in higher dimensions almost surely reside on the boundary of the distribution with a similar distance to the mean (the distribution is a better Gaussian fit).

### B.2  The effects of increasing the projection dimensions

The Gram matrix of the projected vectors can be obtained by considering the inner product of any two vectors $\mathbf{f}, \mathbf{f}'$. As presented in Eqn. (3), this is derived as:

$$
\begin{aligned}
\mathbb{E}_\mathbf{W}\left[(\mathbf{W}^\top\mathbf{f})^\top(\mathbf{W}^\top\mathbf{f}')\right] &= \mathbb{E}_\mathbf{W}\left[(\mathbf{f}^\top\mathbf{W})(\mathbf{W}^\top\mathbf{f}')\right] \\
&= \mathbb{E}_\mathbf{W}\left[\sum_i^M \mathbf{W}_{(i)}^2\mathbf{f}^\top\mathbf{f}' + \sum_{i\ne j}^M \mathbf{W}_{(i)}^\top\mathbf{W}_{(j)}\mathbf{f}^\top\mathbf{f}'\right] \\
&= \underbrace{\sum_i^M \mathbb{E}_\mathbf{W}\left[\mathbf{W}_{(i)}^2\right]\mathbf{f}^\top\mathbf{f}'}_{=M\sigma^2\mathbf{f}^\top\mathbf{f}'} + \underbrace{\sum_{i\ne j}^M \mathbb{E}_\mathbf{W}\left[\mathbf{W}_{(i)}\right]^\top\mathbb{E}_\mathbf{W}\left[\mathbf{W}_{(j)}\right]\mathbf{f}^\top\mathbf{f}'}_{=0}, \quad (7)
\end{aligned}
$$

where the second term is zero for any two zero-mean independently drawn random vectors $\mathbf{W}_{(i)}, \mathbf{W}_{(j)}$. We can derive the following from this expansion:

1. Using the Chernoff inequality, for any two vectors, we have

$$\mathbb{P}\left(\left|(\mathbf{W}^\top\mathbf{f})^\top(\mathbf{W}^\top\mathbf{f}') - \mathbb{E}_\mathbf{W}\left[(\mathbf{W}^\top\mathbf{f})^\top(\mathbf{W}^\top\mathbf{f}')\right]\right| > \epsilon'M\sigma^2\right) \le 2\exp\left(-\frac{\epsilon'^2M\sigma^2}{(2+\epsilon')}\right)$$

$$\mathbb{P}\left(\left|(\mathbf{W}^\top\mathbf{f})^\top(\mathbf{W}^\top\mathbf{f}') - M\sigma^2\mathbf{f}^\top\mathbf{f}'\right| > \epsilon'M\sigma^2\right) \le 2\exp\left(-\frac{\epsilon'^2M\sigma^2}{(2+\epsilon')}\right)$$

$$\mathbb{P}\left(\left|\frac{(\mathbf{W}^\top\mathbf{f})^\top(\mathbf{W}^\top\mathbf{f}')}{M\sigma^2} - \mathbf{f}^\top\mathbf{f}'\right| > \epsilon'\right) \le 2\exp\left(-\frac{\epsilon'^2M\sigma^2}{(2+\epsilon')}\right). \tag{8}$$

This bound indicates that as the dimension of the projections increases, it is more likely for the inner product of any two vectors and their projections to be distinct. In other words, it is increasingly unlikely for the inner product of two vectors and their projections to be equal as $M$ increases.

2. As $M$ increases, it is more likely for the inner product of any two randomly projected instances to be distinct (i.e. the inner products in the projected space are more likely to be larger than some constant). That is because, using Markov's inequality, for a larger $M$ it is easier to choose larger $\epsilon_2$ that satisfies

$$\mathbb{P}\left(|(\mathbf{W}^\top\mathbf{f})^\top(\mathbf{W}^\top\mathbf{f}')| \ge \epsilon_2\right) \le \frac{M\sigma^2}{\epsilon_2}. \tag{9}$$

## B.3 Connection to least squares

The score of Eqn. (5) can be written in matrix form as

$$\mathbf{y}_{\text{test}} = \mathbf{h}_{\text{test}}(\mathbf{G} + \lambda\mathbf{I})^{-1}\mathbf{C}, \tag{10}$$

where $\mathbf{h}_{\text{test}} := \phi(\mathbf{f}_{\text{test}}^\top\mathbf{W})$ are the randomly projected activations following element-wise nonlinear activation $\phi(\cdot)$. The form $\mathbf{W}_{\text{o}} := (\mathbf{G} + \lambda\mathbf{I})^{-1}\mathbf{C}$ arises in fundamental theory of least squares regression. We can write this as

$$\mathbf{W}_o = (\mathbf{G} + \lambda\mathbf{I})^{-1}\mathbf{H}\mathbf{Y}_{\text{train}} \tag{11}$$
$$= (\mathbf{H}\mathbf{H}^\top + \lambda\mathbf{I})^{-1}\mathbf{H}\mathbf{Y}_{\text{train}}, \tag{12}$$

where $\mathbf{Y}_{\text{train}}$ is a $N \times K$ one-hot encoded target matrix, and $\mathbf{H}$ is as defined in Section 4.3. Eqn. (12) is well known [35, Eqn (7.33), p. 226] to be the minimum mean squared error solution to the $l_2$ regularized set of equations defined by

$$\mathbf{Y}_{\text{train}}^\top = \mathbf{W}_{\text{o}}^\top\mathbf{H}. \tag{13}$$

For parameter $\lambda \geq 0$, this is usually expressed mathematically as

$$\mathbf{W}_{\text{o}} = \text{argmin}_{\mathbf{W}}(||\mathbf{Y}_{\text{train}}^\top - \mathbf{W}^\top\mathbf{H}||_2^2 + \lambda||\mathbf{W}||_2^2). \tag{14}$$

Consequently, when not using CL, optimizing a linear output head using SGD with mean square error loss and weight decay equal to $\lambda$ will produce a loss lower bounded by that achieved by directly computing $\mathbf{W}_{\text{o}}$.

## B.4 Connection to Linear Discriminant Analysis, Mahalanobis distance and ZCA whitening

There are two reasons why we use Eqn. (2) for **Algorithm 1** rather than utilize LDA. First and foremost, is the fact that our formulation is mean-square-error optimal, as per Section B.3. Second, use of the inverted Gram matrix results in two convenient simplifications compared with LDA: (i) the simple form of Eqn. (2) can be used for inference instead of a form in which biases calculated from class-prototypes are required, and (ii) accumulation of updates to the Gram matrix and class-prototypes during the CL process as in Eqn. (4) are more efficient than using a covariance matrix.

We now work through the theory that leads to these conclusions. The insight gained is that they show that using Eqn. (2) is equivalent to learning a linear classifier optimized with a mean square error loss function and $l_2$ regularization, applied to the feature vectors of the training set. These derivations apply in both a CL and non-CL context. For CL, the key is to realise that CPs and second-order statistics can be accumulated identically for the same overall set of data, regardless of the sequence in which it arrives for training.

### B.4.1 Preliminaries: Class-Prototypes

Class-prototypes for CL after task $T$ can be defined as

$$\bar{\mathbf{c}}_y = \frac{1}{n_y}\sum_{t=1}^{T}\sum_{n=1}^{N_t}\mathbf{h}_{t,n}\mathcal{I}_{t,n} \quad y = 1, \ldots, K, \tag{15}$$

where $\mathcal{I}_{t,n}$ is an indicator function with value 1 if the $n$–th training sample in the $t$–th task is in class $y$ and zero otherwise, $n_y$ is the number of training samples in each class and $\mathbf{h}_{t,n}$ is an $M$-dimensional projected and activated feature vector. For RanPAC we drop the mean and use

$$\mathbf{c}_y = \sum_{t=1}^{T}\sum_{n=1}^{N_t}\mathbf{h}_{t,n}\mathcal{I}_{t,n} \quad y = 1, \ldots, K. \tag{16}$$

For Nearest Class Mean (NCM) classifiers, and a cosine similarity metric as used by [65], the score for argmax classification is

$$s_y = \frac{\mathbf{f}_{\text{test}}^\top\bar{\mathbf{c}}_y}{||\mathbf{f}_{\text{test}}|| \cdot ||\bar{\mathbf{c}}_y||}, \quad y = 1, \ldots, K, \tag{17}$$

where the $\bar{\mathbf{c}}_y$ are calculated from feature vectors $\mathbf{f}_{t,n}$ rather than $\mathbf{h}_{t,n}$. These cosine similarities depend only on first order feature statistics, i.e. the mean feature vectors for each class. In contrast, LDA and our approach use second-order statistics i.e. correlations between features within the feature vectors.

### B.4.2 Relationship to LDA and Mahalanobis distance

The form of Eqn. (2) resembles Linear Discriminant Analysis (LDA) classification [35, Eqn. (4.38), p. 104]. For LDA, the score from which the predicted class for $\mathbf{f}_{\text{test}}$ is chosen is commonly expressed in a weighting and bias form

$$
\begin{aligned}
\psi_y &= \mathbf{f}_{\text{test}}\mathbf{a} + \mathbf{b} &\text{(18)}\\
&= \mathbf{f}_{\text{test}}^\top \mathbf{S}^{-1}\bar{\mathbf{c}}_y - 0.5\bar{\mathbf{c}}_y^\top \mathbf{S}^{-1}\bar{\mathbf{c}}_y + \log(\pi_y), \qquad y = 1, \dots, K, &\text{(19)}
\end{aligned}
$$

where $\mathbf{S}$ is the $M \times M$ covariance matrix for the $M$ dimensional feature vectors and $\pi_y$ is the frequency of class $y$.

Finding the maximum $\psi_y$ is equivalent to a minimization involving the Mahalanobis distance, $d_M := \sqrt{(\mathbf{f}_{\text{test}} - \bar{\mathbf{c}}_y)^\top \mathbf{S}^{-1}(\mathbf{f}_{\text{test}} - \bar{\mathbf{c}}_y)}$, between a test vector and the CPs, i.e. minimizing

$$
\begin{aligned}
\hat{\psi}_y &= d_M^2 - \log(\pi_y^2) &\text{(20)}\\
&= (\mathbf{f}_{\text{test}} - \bar{\mathbf{c}}_y)^\top \mathbf{S}^{-1}(\mathbf{f}_{\text{test}} - \bar{\mathbf{c}}_y) - \log(\pi_y^2) \qquad y = 1, \dots .K. &\text{(21)}
\end{aligned}
$$

This form highlights that if all classes are equiprobable, minimizing the Mahalanobis distance suffices for LDA classification.

### B.4.3 Relationship to ZCA whitening

We now consider ZCA whitening [23]. This process linearly transforms data $\mathbf{H}$ to $\mathbf{D}_\mathbf{Z}\mathbf{H}$ using the Mahalanobis transform, $\mathbf{D}_\mathbf{Z} = \mathbf{S}^{-0.5}$. Substituting for $\mathbf{S}$ in Eqn. (21) gives

$$
\begin{aligned}
\hat{\psi}_y &= (\mathbf{D}_\mathbf{Z}(\mathbf{f}_{\text{test}} - \bar{\mathbf{c}}_y))^\top (\mathbf{D}_\mathbf{Z}(\mathbf{f}_{\text{test}} - \bar{\mathbf{c}}_y)) - \log(\pi_y^2) &\text{(22)}\\
&= ||(\mathbf{D}_\mathbf{Z}(\mathbf{f}_{\text{test}} - \bar{\mathbf{c}}_y)||_2^2 - \log(\pi_y^2), \qquad y = 1, \dots, K. &\text{(23)}
\end{aligned}
$$

This form implies that if all classes are equiprobable, LDA classification is equivalent to minimizing Euclidian distance following ZCA whitening of both a test sample vector, and all CPs.

### B.4.4 Decorrelating Class-prototypes

The score in Eqn. (2) can be derived in a manner that highlights the similarities and differences to the Mahalanobis transform. Here we use the randomly projected features $\mathbf{h}$ instead of extracted features $\mathbf{f}$. We choose a linear transform matrix $\mathbf{D} \in \mathbb{R}^{M \times M}$ that converts the Gram matrix, $\mathbf{G} = \mathbf{H}\mathbf{H}^\top$, to the identity matrix, i.e. $\mathbf{D}$ must satisfy

$$
(\mathbf{D}\mathbf{H})(\mathbf{D}\mathbf{H})^\top = \mathbf{I}_{M \times M}. \tag{24}
$$

It is easy to see that $\mathbf{D}^\top \mathbf{D} = (\mathbf{H}\mathbf{H}^\top)^{-1} = \mathbf{G}^{-1}$. Next, treating test samples and class prototypes as originating from the same distribution as $\mathbf{H}$, we can consider the dot products

$$
\begin{aligned}
\hat{s}_y &= (\mathbf{D}\mathbf{h}_{\text{test}})^\top (\mathbf{D}\mathbf{c}_y)\\
&= \mathbf{h}_{\text{test}}^\top \mathbf{G}^{-1}\mathbf{c}_y, \qquad y = 1, \dots, K. &\text{(25)}
\end{aligned}
$$

Hence, the same similarities arise as those derivable from the minimum mean square error formulation of Eqn. (14). When compared with the Mahalanobis transform, $\mathbf{D}_\mathbf{Z} = \mathbf{S}^{-0.5}$, the difference here is that the Gram matrix becomes equal to the identity rather than the covariance matrix.

LDA is the same as our method if all classes are equiprobable and $\mathbf{G} = \mathbf{S}$, which happens if all $M$ features have a mean of zero, which is not true in general.

## Appendix C    Training and implementation

### C.1    Optimizing ridge regression parameter in Algorithm 1

With reference to the final step in **Algorithm 1**, we optimized $\lambda$ as follows. For each task, $t$, in Phase 2, the training data for that task was randomly split in the ratio 80:20. We parameter swept over 17 orders of magnitude, namely $\lambda \in \{10^{-8}, 10^{-7}, \dots, 10^8\}$ and for each value of $\lambda$ used $\mathbf{C}$ and $\mathbf{G}$ updated with only the first 80% of the training data for task $t$ to then calculate $\mathbf{W}_\text{o} = (\mathbf{G} + \lambda\mathbf{I})^{-1}\mathbf{C}$.

We then calculated the mean square error between targets and the set of predictions of the form $\mathbf{h}^\top \mathbf{W}_\mathrm{o}$ for the remaining 20% of the training data. We then chose the value of $\lambda$ that minimized the mean square error on this 20% split. Hence, $\lambda$ is updated after every task, and computed in a manner compatible with CL, i.e. without access to data from previous training tasks. It is worth noting that optimizing $\lambda$ to a value between orders of magnitude will potentially slightly boost accuracies. Note also that choosing $\lambda$ only for data from the current task may not be optimal relative to non-CL learning on the same data, in which case the obvious difference would be to optimize $\lambda$ on a subset of training data from the entire training set.

## C.2   Training details

For Phase 2 in **Algorithm 1**, the training data was used as follows. For each sample, features were extracted from a frozen pretrained model, in order to update the $\mathbf{G}$ and $\mathbf{C}$ matrices. We then computed $\mathbf{W}_\mathrm{o}$ using matrix inversion and multiplication. Hence, no SGD based weight updates are required in Phase 2.

For Phase 1 in **Algorithm 1**, we used SGD to train the parameters of PETL methods, namely AdaptFormer [6], SSF [28], and VPT [21]. For each of these, we used batch sizes of $48$, a learning rate of $0.01$, weight decay of $0.0005$, momentum of $0.9$, and a cosine annealing schedule that finishes with a learning rate of $0$. Generally we trained for 20 epochs, but in some experiments reduced to fewer epochs if overfitting was clear. When using these methods, softmax and cross-entropy loss was used. The number of classes was equal to the number in the first task, i.e. $N_1$. The resulting trained weights and head were discarded prior to commencing Phase 2 of **Algorithm 1**.

For reported data in Table 1 using linear probes we used batch sizes of $128$, a learning rate of $0.01$ in the classification head, weight decay of $0.0005$, momentum of $0.9$ and training for $30$ epochs. For full fine-tuning (Table 2), we used the same settings, but additionally used a learning rate in the body (the pre-trained weights of the ViT backbone) of $0.0001$. We used a fixed learning rate schedule, with no reductions in learning rate. We found for fine-tuning that the learning rate lower in the body than the head was essential for best performance.

Data augmentation during training for all datasets included random resizing then cropping to $224 \times 224$ pixels, and random horizontal flips. For inference, images are resized to short side $256$ and then center-cropped to $224 \times 224$ for all datasets except CIFAR100, which are simply resized from the original $32 \times 32$ to $224 \times 224$.

Given our primary comparison is with results from [65], we use the same seed for our main experiments, i.e. a seed value of 1993. This enables us to obtain identical results as [65] in our ablations. However, we note that we use Average Accuracy as our primary metric, whereas [65] in their public repository calculate overall accuracy after each task which can be slightly different to Average Accuracy. For investigation of variability in Section F, we also use seeds 1994 and 1995.

## C.3   Training and Inference Speed

The speed for inference with RanPAC is negligibly different to the speed of the original pre-trained network, because both the RP layer and the output linear classification head (comprised from decorrelated class prototypes) are implemented as simple fully-connected layers on top of the underlying network. For training, Phase 1 trains PETL parameters using SGD for 20 epochs, on $(1/T)$'th of the training set, so is much faster than joint training. Phase 2 is generally only slightly slower than running all training data through the network in inference mode, because the backbone is frozen. The slowest part is the inversions of the Gram matrix, during selection of $\lambda$, but even for $M = 10000$, this is in the order of 1 minute per task on a CPU, which can be easily optimized further if needed. Our view is that the efficiency and simplicity of our approach compared to the alternatives is very strong.

## C.4   Compute

All experiments were conducted on a single PC running Ubuntu 22.04.2 LTS, with 32 GB of RAM, and Intel Core i9-13900KF x32 processor. Acceleration was provided by a single NVIDIA GeForce 4090 GPU.

## Appendix D   Parameter-Efficient Transfer Learning (PETL) methods

We experiment with the same three methods as [65], i.e. AdaptFormer [6], SSF [28], and VPT [21]. Details can be found in [65]. For VPT, we use the deep version, with prompt length 5. For AdaptFormer, we use the same settings as in [65], i.e. with projected dimension equal to 64.

## Appendix E   Datasets

### E.1   Class Incremental Learning (CIL) Datasets

The seven CIL datasets we use are summmarised in Table A2. For Imagenet-A, CUB, Omnibenchmark and VTAB, we used specific train-validation splits defined and outlined in detail by [65]. Those four datasets, plus Imagenet-R (created by [56]) were downloaded from links provided at `https://github.com/zhoudw-zdw/RevisitingCIL`. CIFAR100 was accessed through torchvision. Stanford cars was downloaded from `https://ai.stanford.edu/~jkrause/cars/car_dataset.html`.

For Stanford Cars and $T = 10$, we use 16 classes in $t = 1$ and 20 in the 9 subsequent tasks. It is interesting to note that VTAB has characteristics of both CIL and DIL. Unlike the three DIL datasets we use, VTAB introduces new disjoint sets of classes in each of 5 tasks that originate in different domains. For this reason we use only $T = 5$ for VTAB, whereas we explore $T = 5$, $T = 10$ and $T = 20$ for the other CIL datasets.

|  | Original | CL version | $N$ | # val samples | $K$ |
|---|---|---|---|---|---|
| CIFAR100 | [26] | [42] | 50000 | 10000 | 100 |
| Imagenet-R | [13] | [56] | 24000 | 6000 | 200 |
| Imagenet-A | [14] | [65] | 5981 | 5985 | 200 |
| CUB | [52] | [65] | 9430 | 2358 | 200 |
| OmniBenchmark | [64] | [65] | 89697 | 5985 | 300 |
| VTAB | [62] | [65] | 1796 | 8619 | 50 |
| Stanford Cars | [25] | [63] | 8144 | 8041 | 196 |

Table A2: **CIL Datasets.** We list references for the original source of each dataset and for split CL versions of them. In the column headers, $N$ is the total number of training samples, $K$ is the number of classes following training on all tasks, and # val samples is the number of validation samples in the standard validation sets.

### E.2   Domain Incremental Learning (DIL) Datasets

For DIL, we list the domains for each dataset in Table A3. Further details can be found in the cited references in the first column of Table A3. As in previous work, validation data includes samples from each domain for CDDB-Hard, and DomainNet, but three entire domains are reserved for CORe50.

## Appendix F   Additional results

### F.1   Preliminaries

We provide results measured by Average Accuracy and Average Forgetting. Average Accuracy is defined as [30]

$$A_t = \frac{1}{t} \sum_{i=1}^{t} R_{t,i}, \tag{26}$$

where $R_{t,i}$ are classification accuracies on the $i$–th task, following training on the $t$-th task. Average Forgetting is defined as [3]

$$F_t = \frac{1}{t-1} \sum_{i=1}^{t-1} \max_{t' \in \{1,2,\dots,t-1\}} (R_{t',i} - R_{t,i}). \tag{27}$$

|  | $K$ | $T$ | $N$ | $N_t, t = 1, \ldots, T$ | $N_{\mathrm{val}}$ | val set |
|---|---|---|---|---|---|---|
| CORe50 [29] | 50 | 8 | 119894 |  | 44972 | S3, S7, S10 |
| (Nearly class |  |  |  | $N_1 = 14989$ (S1) |  |  |
| balanced) |  |  |  | $N_2 = 14986$ (S2) |  |  |
| ($\sim 2400$ / class) |  |  |  | $N_3 = 14995$ (S4) |  |  |
|  |  |  |  | $N_4 = 14966$ (S5) |  |  |
|  |  |  |  | $N_5 = 14989$ (S6) |  |  |
|  |  |  |  | $N_6 = 14984$ (S8) |  |  |
|  |  |  |  | $N_7 = 14994$ (S9) |  |  |
|  |  |  |  | $N_8 = 14991$ (S11) |  |  |
| CDDB-Hard [27] | 2 | 5 | 16068 |  | 5353 | Standard |
| (Class balanced |  |  |  | $N_1 = 6000$ (gaugan) | 2000 |  |
| both train and val) |  |  |  | $N_2 = 2400$ (biggan) | 800 |  |
|  |  |  |  | $N_3 = 6208$ (wild) | 2063 |  |
|  |  |  |  | $N_4 = 1200$ (whichfaceisreal) | 400 |  |
|  |  |  |  | $N_5 = 260$ (san) | 90 |  |
| DomainNet [40] | 345 | 6 | 409832 |  | 176743 | Standard |
| (Imbalanced) |  |  |  | $N_1 = 120906$ (Real) | 52041 |  |
|  |  |  |  | $N_2 = 120750$ (Quickdraw) | 51750 |  |
|  |  |  |  | $N_3 = 50416$ (Painting) | 21850 |  |
|  |  |  |  | $N_4 = 48212$ (Sketch) | 20916 |  |
|  |  |  |  | $N_5 = 36023$ (Infograph) | 15582 |  |
|  |  |  |  | $N_6 = 33525$ (Clipart) | 14604 |  |

Table A3: **DIL Datasets.** $K$ is the number of classes, all of which are included in each task. $T$ is the total number of tasks, $N$ is the total number of training samples across all tasks and $N_{\mathrm{val}}$ is the number of validation samples, either overall (first row per dataset) or per task. Also shown are the number of training samples in each task, $N_t$, and the domain names for the corresponding tasks. Core50 was downloaded from `http://bias.csr.unibo.it/maltoni/download/core50/core50_imgs.npz`. CDDB was downloaded from `https://coral79.github.io/CDDB_web/`. DomainNet was downloaded from `http://ai.bu.edu/M3SDA/#dataset` – we used the version labelled as "cleaned version, recommended."

Note that for CIL, we calculate the $R_{t,i}$ as the accuracy for the subset of classes in $\mathcal{D}_i$.

For DIL, since all classes are present in each task, $R_{t,i}$ has a different nature, and is dependent on dataset conventions. For CORe50, the validation set consists of three domains not used during training (S3, S7 and S10, as per Table A3), and in this case, each $R_{t,i}$ is calculated on the entire validation set. Therefore $R_{t,i}$ is constant for all $i$ and $A_t = R_{t,0}$. For CDDB-Hard and DomainNet, for the results in Table 3 we treated the validation data in the same way as for CORe50. However, it is also interesting to calculate accuracies for validation subsets in each domain – we provide such results in the following subsection.

### F.2  Variability and performance on each task

Fig. A2 shows, for three different random seeds, Average Accuracies and Average Forgetting after each CIL task matching Table 1 in Section 5, without PETL (Phase 1). Due to not using PETL, the only random variability is (i) the choice of which classes are randomly assigned to which task and (ii) the random values sampled for the weights in $\mathbf{W}$. We find that after the final task, the Average Accuracy is identical for each random seed when all classes have the same number of samples, and are nearly identical otherwise. Clearly the randomness in $\mathbf{W}$ has negligible impact. For all datasets except VTAB, the value of RP is clear, i.e. RP (black traces) delivers higher accuracies by the time of the final task than not using RP, or when only using NCM (blue traces). The benefits of second-order statistics even without RP are also evident (magenta traces), with Average Accuracies better than NCM. Note that VTAB always has the same classes assigned to the same tasks, which is why only one repetition is shown. The difference with VTAB is also evident in the average accuracy trend as the number of tasks increases, i.e. the Average Accuracy does not have a clearly decreasing trend as the number of tasks increases. This is possibly due to the difference in domains for each Task making it less likely that confusions occur for classes in one task against those in another task.

Fig. A3 shows comparisons when Phase 1 (PETL) is used. The same trends are apparent, except that due to the SGD training required for PETL parameters, greater variability in Average Accuracy after the final task can be seen. Nevertheless, the benefits of using RP are clearly evident.

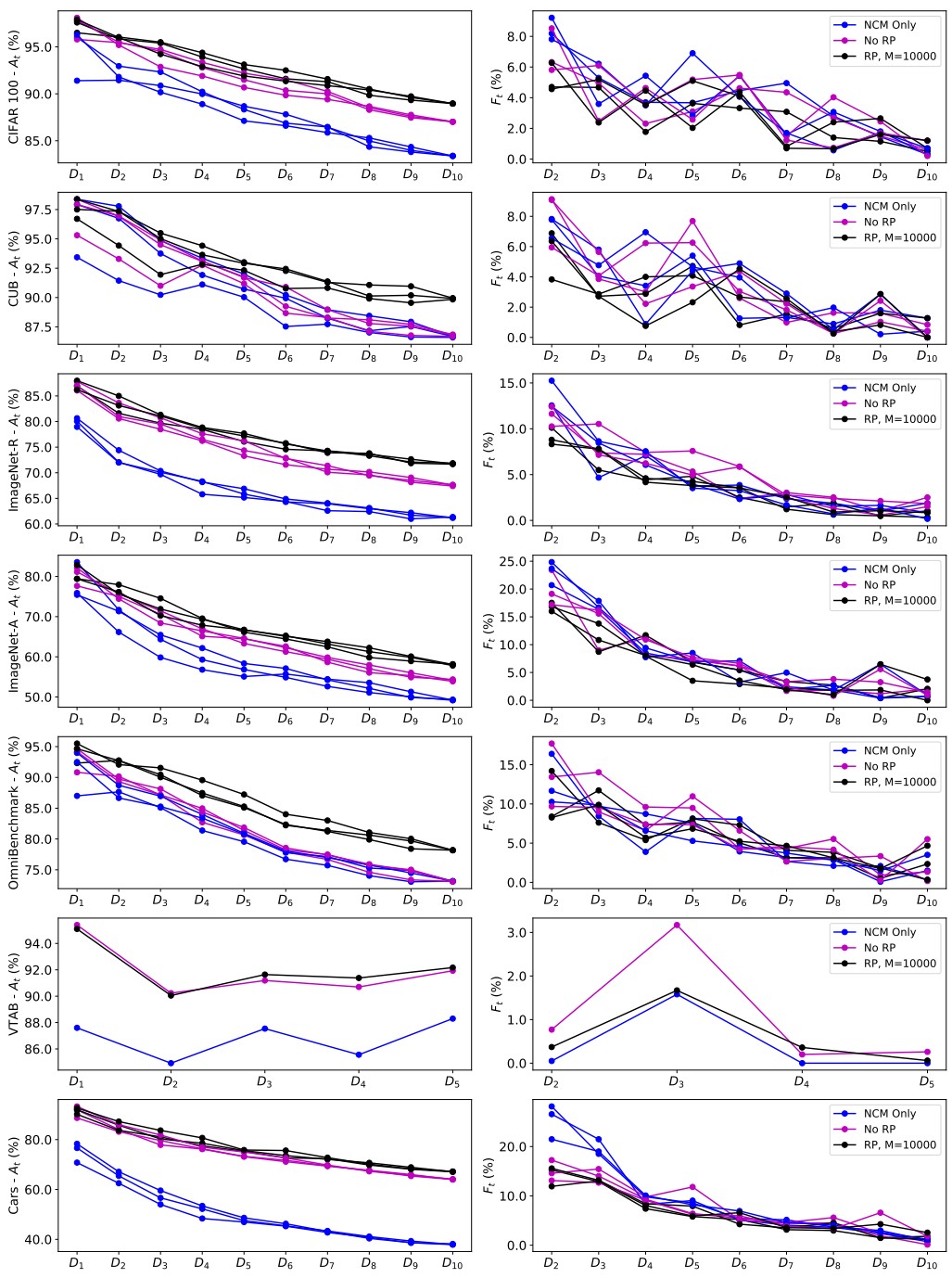

Figure A2: **Average Accuracies and Forgetting after each Task for CIL datasets (no Phase 1).** The left column shows Average Accuracies for three random seeds after each of $T$ tasks for the seven CIL datasets, without using Phase 1. The right column shows Average Forgetting.

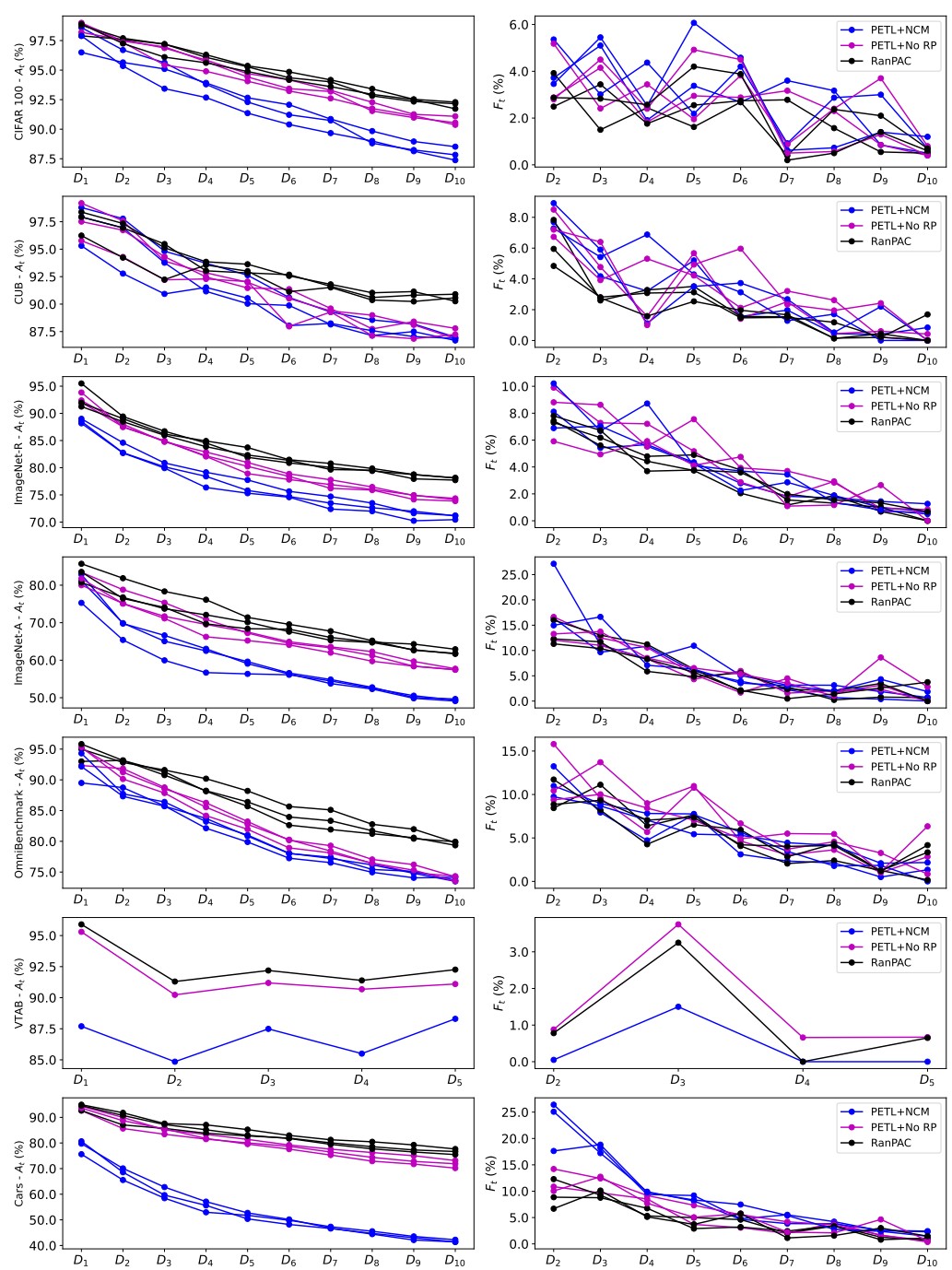

Figure A3: **Average Accuracies and Forgetting after each Task for CIL datasets (with Phase 1).** The left column shows Average Accuracies for three random seeds after each of $T$ tasks for the seven CIL datasets, for the best choice of PETL method in Phase 1. The right column shows Average Forgetting.

Fig. A4 shows how accuracy on individual domains changes as tasks are added through training for the DIL dataset, CDDB-Hard. The figure shows that after training data for a particular domain is first used, that accuracy on the corresponding validation data for that domain tends to increase. In some cases forgetting is evident, e.g. for the 'wild' domain, after training on $T_4$ and $T_5$. The figure also shows that averaging accuracies for individual domains ('mean over domains') is significantly lower than 'Overall accuracy'. For this particular dataset, 'Average accuracy' is potentially a misleading metric as it does not take into account the much lower number of validation samples in some domains, e.g. even though performance on 'san' increases after training on it, it is still under 60%, which is poor for a binary classification task.

Fig. A5 shows the same accuracies by domain for DomainNet. Interestingly, unlike CDDB-Hard, accuracy generally increases for each Domain as new tasks are learned. This suggests that the pre-trained model is highly capable of forming similar feature representation for the different domains in DomainNet, such that increasing amounts of training data makes it easier to discriminate between classes. The possible difference with CDDB-Hard is that that dataset is a binary classification task in which discriminating between the 'real' and 'fake' classes is inherently more difficult and not reflected in the data used to train the pre-trained model.

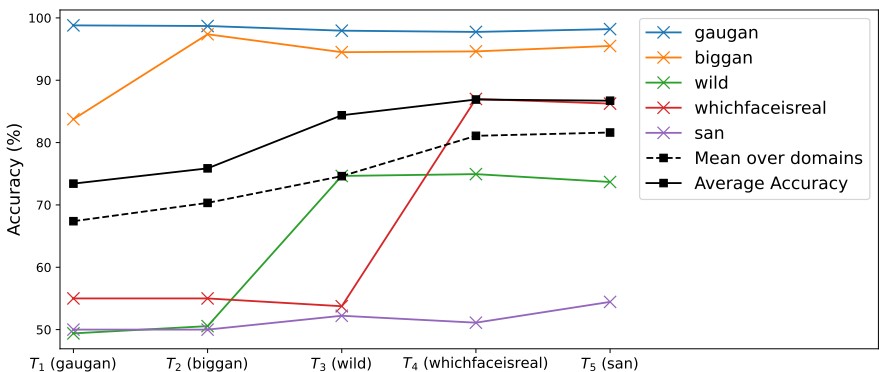

Figure A4: **Accuracies for DIL dataset CDDB-Hard.** Results are shown for the full RanPAC algorithm, using VPT as the PETL method, and $M = 10000$. Each task, $T_i$ corresponds to learning on a new domain as shown on the x-axis. The accuracies shown in colors are those for individual domains, following training on the domain shown on the x-axis. 'Mean over domains' is the average of the five domain accuracies after each task.

### F.3 Comparison of impact of Phase 1 for different numbers of CIL tasks

As shown in Fig. A2, when Phase 1 is excluded, the *final* Average Accuracy after the final task has negligible variability despite different random assignments of classes to tasks. This is a consequence primarily of Eqns. (4) being invariant to the order in which a completed set of data from all $T$ tasks is used in **Algorithm 1**. As mentioned, the influence of different random values in **W** is negligible. The same effect occurs if the data is split to different numbers of tasks, e.g. $T = 5$, $T = 10$ and $T = 20$, in the event that all classes have equal number of samples, such as CIFAR100.

Therefore, in this section we show in Table A4 results for RanPAC only for the case where variability has greater effect, i.e. when Phase 1 is included, and AdaptMLP chosen. VTAB is excluded from this analysis, since it is a special case where it makes sense to consider only the case of $T = 5$. The comparison results for L2P, DualPrompt and ADaM are copied from [65].

Performance when AdaptMLP is used tends to be better for $T = 5$ and worse for $T = 20$. This is consistent with the fact that more classes are used for first-session training with the PETL method when $T = 5$. By comparison, $T = 20$ is often on par with not using the PETL method at all, indicating that the first session strategy may have little value if insufficient data diversity is available within the first task.

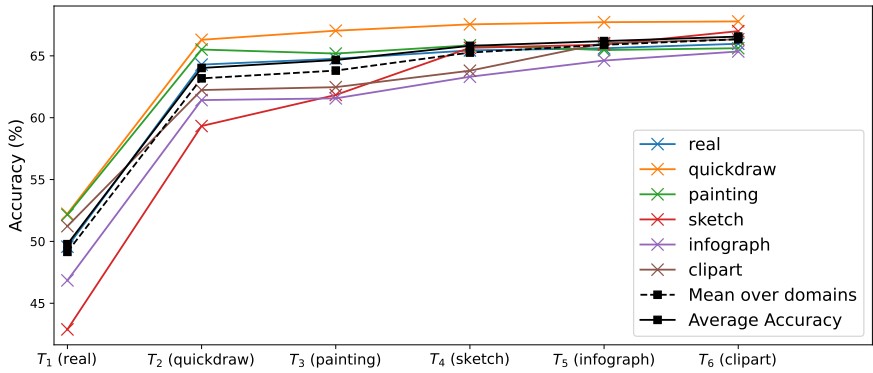

Figure A5: **Accuracies for DIL dataset DomainNet.** Results are shown for the full RanPAC algorithm, using VPT as the PETL method, and $M = 10000$. Each task, $T_i$, corresponds to learning on a new domain as shown on the x-axis. The accuracies shown in colors are those for individual domains, following training on the domain shown on the x-axis. 'Mean over domains' is the average of the six domain accuracies after each task.

| | | $T = 5$ | $T = 10$ | $T = 20$ | RanPAC – No Phase 1 |
|---|---|---|---|---|---|
| CIFAR100 | RanPAC | 92.4% | 92.2% | 90.8% | 89.0% |
| | AdaM | 88.5% | 87.5% | 85.2% | |
| | DualPrompt | 86.9% | 84.1% | 81.2% | |
| | L2P | 87.0% | 84.6% | 79.9% | |
| | NCM | 88.6% | 87.8% | 86.6% | 83.4% |
| ImageNet-R | RanPAC | 79.9% | 77.9% | 74.5% | 71.8% |
| | AdaM | 74.3% | 72.9% | 70.5% | |
| | DualPrompt | 72.3% | 71.0% | 68.6% | |
| | L2P | 73.6% | 72.4% | 69.3% | |
| | NCM | 74.0% | 71.2% | 64.7% | 61.2% |
| ImageNet-A | RanPAC | 63.0% | 58.6% | 58.9% | 58.2% |
| | AdaM | 56.1% | 54.0% | 51.5% | |
| | DualPrompt | 46.6% | 45.4% | 42.7% | |
| | L2P | 45.7% | 42.5% | 38.5% | |
| | NCM | 54.8% | 49.7% | 49.3% | 49.3% |
| CUB | RanPAC | 90.6% | 90.3% | 89.7% | 89.9% |
| | AdaM | 87.3% | 87.1% | 86.7% | |
| | DualPrompt | 73.7% | 68.5% | 66.5% | |
| | L2P | 69.7% | 65.2% | 56.3% | |
| | NCM | 87.0% | 87.0% | 86.9% | 86.7% |
| OmniBenchmark | RanPAC | 79.6% | 79.9% | 79.4% | 78.2% |
| | AdaM | 75.0%* | 74.5% | 73.5% | |
| | DualPrompt | 69.4%* | 65.5% | 64.4% | |
| | L2P | 67.1%* | 64.7% | 60.2% | |
| | NCM | 75.1% | 74.2% | 73.0% | 73.2% |
| Cars | RanPAC | 69.6% | 67.4% | 67.2% | 67.1% |
| | NCM | 41.0% | 38.0% | 37.9% | 37.9% |

Table A4: **Comparison of CIL results for different number of tasks.** For this table, the AdaptMLP PETL method was used for RanPAC. The comparison results for L2P, DualPrompt and ADaM are copied from [65]; for ADaM, the best performing PETL method was used. In most cases, Average Accuracy decreases as $T$ increases, with $T = 20$ typically not significantly better than the case of no PETL ("No Phase 1"). Values marked with * indicates data is for $T = 6$ tasks, instead of $T = 5$. Note that accuracies for comparison methods for Cars were not available from [65]

## F.4    Task Agnostic Continual Learning

Unlike CIL and DIL, 'task agnostic' CL is a scenario where there is no clear concept of a 'task' during training [61]. The concept is also known as 'task-free' [44]. It contrasts with standard CIL where although inference is task agnostic, training is applied to disjoint sets of classes, described as tasks. To illustrate the flexibility of RanPAC, we show here that it is simple to apply it to task agnostic continual learning. We use the Gaussian scheduled CIFAR100 protocol of [57], which was adapted from [44]. We use 200 'micro-tasks' which sample from a gradually shifting subset of classes, with 5 batches of 48 samples in each micro-task. There are different possible choices for how to apply **Algorithm 1**. For instance, the 'first session' for Phase 1 could be defined as a particular total number of samples trained on, e.g. 10% of the anticipated total number of samples. Then in Phase 2, the outer for loop over tasks could be replaced by a loop over all batches, or removed entirely. In both cases, the result for **G** and **C** will be unaffected. The greater challenge is in determining $\lambda$, but generally for a large number of samples, $\lambda$ can be small, or zero. For a small number of training samples, a queue of samples could be maintained such that the oldest sample in the queue is used to update **G** and **C**, with all newer samples used to compute $\lambda$ if inference is required, and then all samples in the buffer added to **G** and **C**.

Here, for simplicity we illustrate application to Gaussian-scheduled CIFAR100 without any Phase 1. Fig. A6 shows how test accuracy changes through training both with and without RP. The green trace illustrates how the number of classes seen in at least one sample increases gradually through training, instead of in a steps like in CIL. The red traces show validation accuracy on the entirety of the validation set. As expected, this increases as training becomes exposed to more classes. The black traces show the accuracy on only the classes seen so far through training. By the end of training, the red and black traces converge, as is expected. Fluctuations in black traces can be partially attributed to not optimizing $\lambda$. The final accuracies with and without RP match the values for $T = 10$ CIL shown in Table 1.

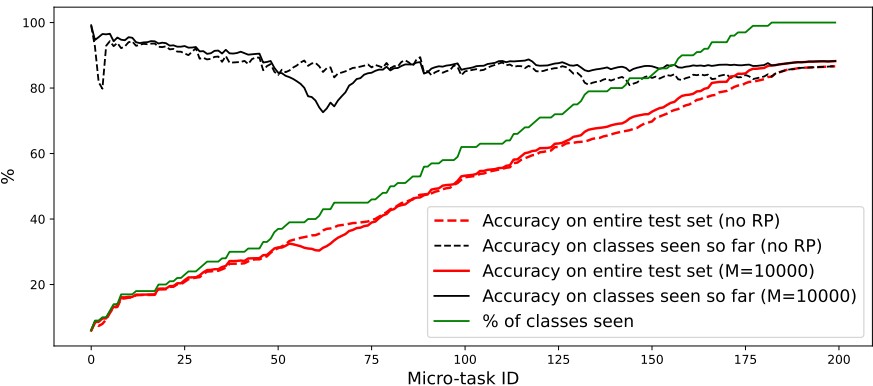

Figure A6: **Task agnostic example**. Application of Phase 2 of **Algorithm 1** to the Gaussian-scheduled CIFAR100 task-agnostic CL protocol.

## F.5    Scaling with projection size

Table A5 shows, for the example of split CIFAR100, that it is important to ensure $M$ is sufficiently large. In order to surpass the accuracy obtained without RP (see 'No RPs or Phase 1' in Table 1), $M$ needs to be larger than 1250.

## F.6    Comparison of PETL Methods and ViT-B/16 backbones

Fig. A7 shows how performance varies with PETL method and by ViT-B/16 backbone. For some datasets, there is a clearly superior PETL method. For example, for CIFAR100, AdaptMLP gives better results than SSF or VPT, for both backbones, for Cars VPT is best, and for ImageNet-A, SSF

| $M$ | Accuracy |
|------|----------|
| 100 | 71.6% |
| 200 | 80.3% |
| 400 | 83.9% |
| 800 | 86.2% |
| 1250 | 86.8% |
| 2500 | 87.7% |
| 5000 | 88.4% |
| 10000 | 88.8% |
| 15000 | 89.0% |

Table A5: **Scaling with $M$ for split CIFAR100.** The table shows final average accuracy for $T = 10$ and no Phase 1, with $\lambda = 100$ as a constant for all $M$, using ViT-B/16 trained on ImageNet21K.

is best. There is also an interesting outlier for VTAB, where VPT and the ImageNet-21K backbone failed badly. This variability by PETL method suggests that the choice of method for first-session CL strategies should be investigated in depth in future work.

Fig. A7 also makes it clear that the same backbone is not optimal for all datasets. For example, the ViT-B/16 model fine-tuned on ImageNet-1K is best for ImageNet-A, but the ViT-B/16 model trained on ImageNet-21K is best for CIFAR100 and OmniBenchmark. For Cars, the best backbone depends on the PETL method.

Fig. A8 summarises how the two ViT networks compare. For all datasets and method variants we plot the Average Accuracy after the final task for one pre-trained ViT network (self-supervised on ImageNet-21K) against the other (fine-tuned on ImageNet-1K). Consistent with Fig. A7, the best choice of backbone is both dataset dependent and method-dependent.

### F.7 Experiments with ResNets

Unlike prompting strategies, our approach works with any feature extractor, e.g. both pre-trained transformer networks and pre-trained convolutional neural networks. To illustrate this, Table A6 and Table A7 shows CIL results for ResNet50 and ResNet 152, respectively, pre-trained on ImageNet. We used $T = 10$ tasks (except for VTAB, which is $T = 5$ tasks). Although this is different to the $T = 20$ tasks used for ResNets by [65], the accuracies we report for NCM are very comparable to those in [65]. As with results for pre-trained ViT-B/16 networks, the use of random projections and second-order statistics both provide significant performance boosts compared with NCM alone. We do not use Phase 1 of **Algorithm 1** here but as shown by [37, 65], this is feasible for diverse PETL methods for convolutional neural networks. Interestingly, ResNet152 with RP produces reduced accuracies compared to ResNet50 on CUB, Omnibenchmark and VTAB. It is possible that this would be remedied by seeking an optimal value of $M$, whereas for simplicity we chose $M = 10000$. Note that unlike the pre-trained ViT-B/16 model, the pre-trained ResNets require preprocessing to normalize the input images.

| Method | CIFAR100 | IN-R | IN-A | CUB | OB | VTAB | Cars |
|--------|----------|------|------|-----|-----|------|------|
| Phase 2 | 75.5% | 56.5% | 35.5% | 70.9% | 68.0% | 88.6% | 48.5% |
| Phase 2 (No RPs) | 73.3% | 54.8% | 36.6% | 62.7% | 60.2% | 87.7% | 42.6% |
| NCM | 61.5% | 43.1% | 25.8% | 55.4% | 57.4% | 82.2% | 26.3% |

Table A6: **Results for ResNet50.** Results are final Average Accuracies for $T = 10$ tasks, except VTAB which is $T = 5$ tasks. Pre-trained weights were IMAGENET1K_V2 variants from PyTorch.

| Method | CIFAR100 | IN-R | IN-A | CUB | OB | VTAB | Cars |
|--------|----------|------|------|-----|-----|------|------|
| Phase 2 | 79.7% | 58.7% | 40.9% | 66.7% | 65.8% | 90.0% | 45.1% |
| Phase 2 (No RPs) | 77.8% | 56.9% | 39.9% | 59.4% | 58.0% | 87.9% | 38.8% |
| NCM | 70.1% | 47.5% | 32.1% | 50.6% | 56.1% | 82.6% | 25.2% |

Table A7: **Results for ResNet152.** Other caption information is the same as Table A6.

ResNet results for DIL are provided in Table A8.

| Method | Backbone | **CORe50** | **CDDB-Hard** | **DomainNet** |
|---|---|---|---|---|
| Phase 2 | ResNet50 | 88.5% | 75.9% | 52.7% |
| Phase 2 (No RPs) | ResNet50 | 86.7% | 75.6% | 41.9% |
| NCM | ResNet50 | 73.1% | 65.1% | 32.9% |
| Phase 2 | ResNet152 | 87.1% | 71.3% | 53.8% |
| Phase 2 (No RPs) | ResNet152 | 86.8% | 71.9% | 42.9% |
| NCM | ResNet152 | 72.0% | 65.6% | 35.1% |

Table A8: **DIL Results for ResNet50 and ResNet152.** Pre-trained weights were IMAGENET1K_V2 variants from PyTorch.

## F.8  Experiments with CLIP vision model

To further verify the general applicability of our method, we show in Table A9 CIL results from using a CLIP [41] vision model as the backbone pre-trained model. The same general trend as for pre-trained ViT-B/16 models and ResNets can be seen, where use of RPs (Phase 2 in **Algorithm 1**) produces better accuracies than NCM alone. Interestingly, the results for Cars is substantially better with the CLIP vision backbone than for ViT-B/16 networks. It is possible that data from a very similar domain as the Cars dataset was included in training of the CLIP vision model. The CLIP result for Phase 2 only (see ablations in Table 1) is also better than ViT/B-16 for Imagenet-R , but for all other datasets, ViT/B-16 has higher accuracy. Note that unlike the pre-trained ViT-B/16 model, the pre-trained CLIP vision model requires preprocessing to normalize the input images.

| Method | **CIFAR100** | **IN-R** | **IN-A** | **CUB** | **OB** | **VTAB** | **Cars** |
|---|---|---|---|---|---|---|---|
| Phase 2 | 85.0% | 76.1% | 39.7% | 79.4% | 75.1% | 89.6% | 90.4% |
| Phase 2 (No RPs) | 80.9% | 67.0% | 36.8% | 72.8% | 66.3% | 89.8% | 85.8% |
| NCM | 77.1% | 68.0% | 32.5% | 73.7% | 67.8% | 79.4% | 85.6% |

Table A9: **Results for CLIP-ViT-B/16.** Results are final Average Accuracies for $T = 10$ tasks, except VTAB which is $T = 5$ tasks. We used pre-trained weights from OpenCLIP, finetuned on ImageNet-1K [19]. The weights were loaded from the `timm` library using the model file name `vit_base_patch32_224_clip_laion2b`.

CLIP results for DIL are provided in Table A10. The CLIP result outperforms that for Phase 2 only for the ViT-B/16 model for CDDB-Hard and DomainNet, but not CORe50 (Table 3).

| Method | **CORe50** | **CDDB-Hard** | **DomainNet** |
|---|---|---|---|
| Phase 2 | 92.9% | 84.4% | 68.7% |
| Phase 2 (No RPs) | 90.2% | 81.7% | 59.1% |
| NCM | 83.9% | 66.9% | 51.0% |

Table A10: **DIL Results for ResNet50 and ResNet152.** Pre-trained weights were IMA-GENET1K_V2 variants from PyTorch.

## F.9  Experiments with regression targets using CLIP vision and language models

Until this point, we have defined the matrix $\mathbf{C}$ as containing Class-Prototypes (CPs), i.e. $\mathbf{C}$ has $N$ columns representing averaged feature vectors of length $M$. However, with reference to Eqn. (13), the assumed targets for regression, $\mathbf{Y}_{\text{train}}$, can be replaced by different targets. Here, we illustrate this using CLIP language model representations as targets, using OpenAI's CLIP ViT-B/16 model.

Using CIFAR100 as our example, we randomly project CLIP's length-512 vision model representations as usual, but also use the length-512 language model's representations of the 100 class names, averaged over templates as in [41]. We create a target matrix of size $N \times 512$ in which each row is the language model's length 512 representation for the class of each sample. We then solve for $\mathbf{W}_{\text{o}} \in \mathcal{R}^{M \times 512}$ using this target instead of $\mathbf{Y}_{\text{train}}$.

When the resulting $\mathbf{W}_o$ is applied to a test sample, the result is a length-512 prediction of a language model representation. In order to translate this to a class prediction, we then apply CLIP in a standard zero-shot manner, i.e. we calculate the cosine similarity between the predictions and each of the normalized language model's length 512 representation for the class of each sample.

The resulting final Average Accuracy for $M = 5000$ and $T = 10$ is $77.5\%$. In comparison, CLIP's zero shot accuracy for the same data is $68.6\%$, which highlights there is value in using the training data to modify the vision model's outputs. When RP is not used, the resulting final Average Accuracy is $71.4\%$.

In future work, we will investigate whether these preliminary results from applying **Algorithm 1** to a combination of pre-trained vision and language models can translate into demonstrable benefits for continual learning.

### F.10 Note on reproducibility

Accuracies reported in Tables 1-3 have been updated since the original submission to reflect those published in our code repository at `https://github.com/zhoudw-zdw/RevisitingCIL`. Note that accuracy results achieved can vary by $\sim \pm 1\%$ for different random seeds, even for the same class ordering, especially for PETL methods.

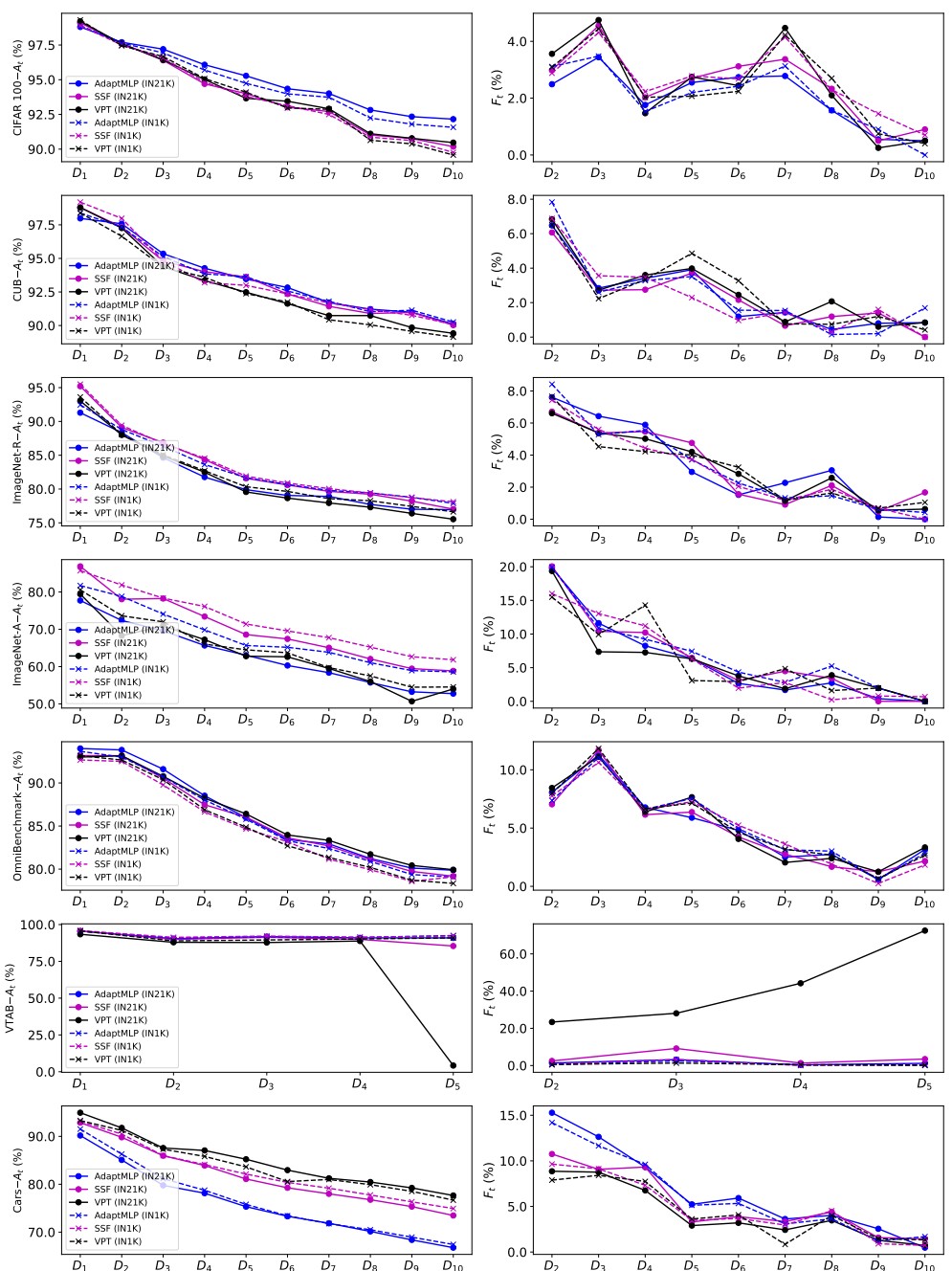

Figure A7: **Comparison of PETL Methods and two ViT models**. For the seven CIL datasets, and $T = 10$, the figure shows Average Accuracy and Average Forgetting after each task, for each of AdaptMLP, SSF and VPT. It shows this information for the two ViT-B/16 pre-trained backbones we primarily investigated.

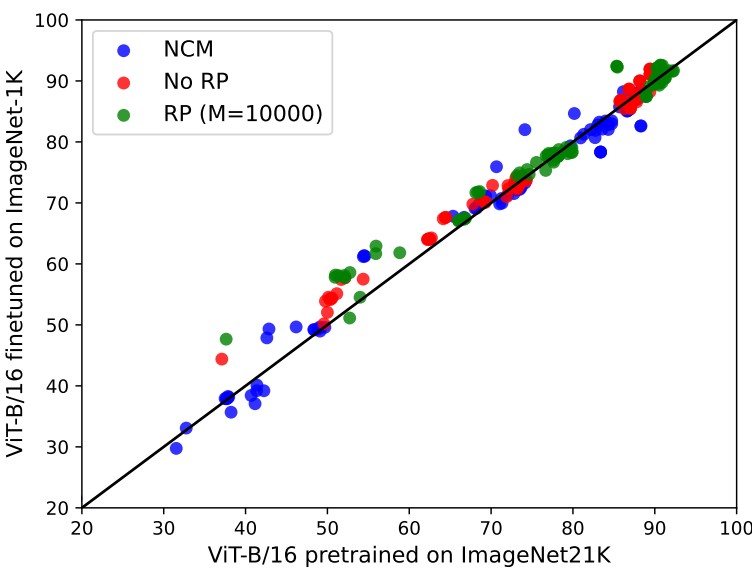

Figure A8: **Comparison of backbone ViTs.** Scatter plot of results for ViT-B/16 models pre-trained on ImageNet1K vs ImageNet21K.

