# OpenReview forum: "RanPAC: Random Projections and Pre-trained Models for Continual Learning"
_NeurIPS.cc/2023/Conference — NeurIPS 2023 poster_

### Official Review · Reviewer_b1mf · 2023-06-08

**Soundness:** 3 good
**Presentation:** 4 excellent
**Contribution:** 3 good
**Rating:** 5
**Confidence:** 5

**Summary:**

This paper proposed a frozen random projection layer with nonlinear activation to exploit pre-trained representations for continual learning. Combining PETL techniques and class prototypes, the proposed method achieves strong performance in class- and domain-incremental learning.

**Strengths:**

1. The paper is well organized and easy to follow.
2. The introduction clearly summarizes the major strategies of leveraging pre-training for continual learning.
3. The proposed method seems to be easy to implement and achieves strong performance.
4. Many recent methods on arxiv have been compared in experiments.

**Weaknesses:**

1. The idea of the proposed method is somewhat incremental. The authors claim that they borrow some ideas from PETL methods. Actually, the Phase 1 seems to (almost) inherit the previous works (line 273-284) and constitute half of the performance improvement (Table 1, 3).
2. Meanwhile, the proposed random projection with nonlinear activation seems equivalent to a randomly-initialized MLP layer, which is a widely-used training trick for transformer backbone in NLP as well as in CV.
3. The authors claim that the extra parameter usage is small compared to the ViT-B/16 backbone (10M vs 84M). However, the compared baselines in Table 1, 3 generally require much smaller parameter usage, especially for prompt-based approaches (e.g., L2P and DualPrompt usually require ~0.03M/Task and 0.3M in total). Therefore, the comparison might be somewhat unfair. Also, how is ''10 times the number of trainable parameters'' (line 302) being calculated?

**Questions:**

My major concerns include the novelty and technical contributions of the proposed method, as well as the parameter usage in experiments. Please refer to Weakness.

**Limitations:**

Yes.

---

> ### Author Rebuttal · Authors · 2023-08-10
>
> We would like to thank the reviewer for their time and feedback. We address the points individually and kindly ask the reviewer to increase their score based on our response.
>
> > “The authors claim that they borrow some ideas from PETL methods. Actually, the Phase 1 seems to (almost) inherit the previous works (line 273-284) and constitute half of the performance improvement (Table 1, 3).”
>
> **Response:** We have been explicit in the use of PETL as methods that our approach can benefit from. Our primary contribution is introducing random projections (RP) for continual learning (see lines 92-103, pages 1-2). Therefore, we can either choose to use PETL or not. The second-last row of Tables 1 and 3 list the performance gains due to RP added on top of PETL, and shows at least 10\% improvement in 9 out of 10 datasets, of which 4 cases are over 20\% improved. Also, the final row of Tables 1 and 3 shows the improvement when PETL is absent, and this is also over 10\% in 6 out of 10 datasets, illustrating that the method does not need PETL to be of value.
>
> > “… the proposed random projection with nonlinear activation seems equivalent to a randomly-initialized MLP layer, which is a widely-used training trick for transformer backbone in NLP as well as in CV."
>
> **Response:** Our RP layer's weights are randomly sampled from the specified distribution and *never trained*, which is very important for CL. This is different to random initialization of *trained* MLP layers. For instance, Gani *et al.* (2022) randomly initialised but then trained MLP weights appended to a pretrained NLP transformer (https://arxiv.org/abs/2210.07240). Please refer us to any specific papers that, like us, do leave random weights frozen.
>
> > "The authors claim that the extra parameter usage is small compared to the ViT-B/16 backbone (10M vs 84M). However, the compared baselines in Table 1, 3 generally require much smaller parameter usage..."
>
> **Response:** Our 10M *non-trainable* parameters are not used in Phase 1 and only ever require forward pass and are never back-propagated through during Phase 2 training, so our training is very fast.
>
> >  "how is ''10 times the number of trainable parameters'' (line 302) being calculated?"
>
> **Response:** Without RP, the number of trainable parameters after $K$ classes is $LK$. When using RP, this increases to $MK$. The increase factor is $(M-L)/L$, i.e. ~10 when $M=10000$ and $L=784$ (but note that M is often effective for smaller values).

---

> ### Comment · Reviewer_b1mf · 2023-08-15
> **Reviewer response**
>
> Thank you for replying to my comments and questions. The rebuttal has addressed some of my concerns. However, my major concerns still remain:
>
> 1. Novelty and technical contribution remain ambiguous. I understand the randomly-initialized MLP might be implemented differently from previous work. However, the use of MLP is a common strategy to improve prompt-tuning, and has been introduced to prompt-based continual learning [1]. At the current stage, I cannot determine whether this contribution is sufficiently strong. It will be more informative to provide a more in-depth comparison of different MLP implementations, especially their potential functions for continual learning.
>
> 2. I understand the MLP is not trained. However, the storage overhead of the 10M parameters seems to be much larger than other baselines (e.g., L2P and DualPrompt), which only use 0.3M in total. This may not be a critical issue, but needs to be clarified as a potential limitation.
>
> [1] Progressive prompts: Continual learning for language models. ICLR 2023.

---

> > ### Author Response · Authors · 2023-08-19
> >
> > Thankyou for your time and the additional remarks and suggestions.  If you find our clarifications satisfactory, we kindly ask to consider increasing the score accordingly.
> >
> > > "The use of MLP is a common strategy to improve prompt-tuning, and has been introduced to prompt-based continual learning [1]. At the current stage, I cannot determine whether this contribution is sufficiently strong. It will be more informative to provide a more in-depth comparison of different MLP implementations, especially their potential functions for continual learning."
> >
> > **Response:** Ref [1] and the work it cites learns input prompts for pretrained models and trains MLPs (different for each task)  that reparameterize them. Our approach is very different, as we do not use either prompts or MLPs; instead, as a Class Prototype strategy for CL, we are focusing on how to extract maximum discriminability from the network's output feature representations. This method does not use multiple layers or bottlenecks commonly used in MLPs. We use random projection of features to a higher dimension and show this simple approach outperforms previous state-of-the-art in continual learning for vision tasks.
> >
> >
> > > "...the storage overhead of the 10M parameters seems to be much larger than other baselines (e.g., L2P and DualPrompt), which only use 0.3M in total. This may not be a critical issue, but needs to be clarified as a potential limitation."
> >
> > **Response:** Thank you for the suggestion. We described on lines 306-308 how theoretically the 10M weights can be represented using 32 times fewer bits, by using the bipolar distribution. Also, the RP projection size $M$ can be smaller, signficantly reducing the total weights (Table A5). We will clarify in the camera-ready as suggested.

---

### Official Review · Reviewer_Q6EM · 2023-07-06

**Soundness:** 3 good
**Presentation:** 3 good
**Contribution:** 3 good
**Rating:** 7
**Confidence:** 4

**Summary:**

The manuscript proposes a Continual Learning method called RanPAC, which belongs to the category of class prototype methods. They use a frozen pretrained model to extract feature vectors from the input images and non-linear random projections to project them to a higher dimensional space. During the training on the task sequence, they continuously update the gram matrix obtained from the projected representations and use that matrix to make predictions. Through mathematical proofs, they show that the distribution of the projected features in the higher dimensional space approaches an isotropic Gaussian. The experimental results in the paper show that their method is superior in accuracy to other state-of-the-art Continual Learning techniques and approaches to the upper bound performance.

**Strengths:**

The idea to investigate the role of random projections of feature vectors obtained from large pretrained models is new in Continual Learning research. The experimental evidence that random projections help in obtaining low correlation coefficients between class prototypes is surely a useful contribution.


**Weaknesses:**

I believe the paper has the following minor weaknesses:
- Page 4, section 2.2, rows 145-161: The paper presents the LDA formula and subsequently mentions that the authors employed the Gram matrix. Nevertheless, it was not clear to me the motivation behind this choice.
- Page 5, figure 2: I didn't quite understand the plots in Figure 2, left. They are introduced in Section 2, and it is explained that they represent the similarities of class prototypes. However, the meaning of "true class" and "inter-class" was not very clear to me.
- Page 5-6, sections 3.2 and 3.3: I found the mathematical parts hard to understand. I think it would be more beneficial to provide simplified explanations of the equations, using simpler language to clarify their meaning.

**Questions:**

The contribution of the work and the results show are valid. However, as I said in the previous sections, I believe that some parts of the paper could be improved and explained better.

**Limitations:**

yes

---

> ### Author Rebuttal · Authors · 2023-08-10
>
> We would like to thank the reviewer for their time and feedback. We address the points individually and kindly ask the reviewer to increase their score based on our response.
>
> > "The paper presents the LDA formula and subsequently mentions that the authors employed the Gram matrix. Nevertheless, it was not clear to me the motivation behind this choice."
>
> **Response:** This is explained in detail in the Appendix, Sec.B.4, where 3 reasons are listed. In the final version we will clarify by inserting a summary in the Approach section.
>
> > "...the meaning of "true class" and "inter-class" was not very clear to me."
>
> **Response:** “True class” refers to the histogram of cosine similarities between a sample’s feature vector and the class prototype for the class label corresponding to that sample. “Inter-class” refers to the histogram of cosine similarities between a sample’s feature vector and the class prototypes for the set of N-1 classes not equal to the sample’s class label. We will clarify in the camera-ready.
>
> > "I found the mathematical parts hard to understand. I think it would be more beneficial to provide simplified explanations of the equations, using simpler language to clarify their meaning."
>
> **Response:** Appendix B (Sections B1-B4), expand on the details of all the mathematical concepts.  We will add clarifying remarks to enhance readability in the camera-ready version.

---

> > ### Comment · Reviewer_Q6EM · 2023-08-17
> > **Rebuttal response**
> >
> > I thank the authors for the rebuttal. After consideration, I will maintain my previous score of Accept for the manuscript.

---

### Official Review · Reviewer_2MfY · 2023-07-08

**Soundness:** 3 good
**Presentation:** 3 good
**Contribution:** 3 good
**Rating:** 7
**Confidence:** 4

**Summary:**

The authors propose a method for replay-free continual learning from a pre-trained model based on random projections and prototypes. The method has a high parameter cost compared to SOTA prompting-based methods, but also is a unique method which strongly outperforms these SOTA methods. Overall, the experiments and analysis on class-incremental and domain-incremental learning provide a strong motivation for the proposed approach.

**Strengths:**

1) I enjoyed the writing style of this paper. It was clear and scientific.
2) I appreciate the authors thinking outside of the box. Rather than another prompting method, they propose a clever new direction with high motivation and experimental justification. The method has good intuition and motivation for the problem setting.
3) The experiment section is very comprehensive and mostly satisfying.
4) I explored the SM and am very happy with all of the additional details and experiments provided by the authors.

**Weaknesses:**

1) There seems to be large number of additional trainable parameters. While for 10 tasks, this is only 10/84 of the model size, it could double the necessary parameters for a longer task sequence.
2) Speaking of task sequences, I would like to see how the method performs for longer task sequences. A simple and easy experiment would be 20 task ImageNet-R.
3) Computation time analysis (training and inference) seems to be missing.

**Questions:**

a) How does the method perform on longer task sequences? For example, 20-tasks of ImageNet-R? There is one table in the SM with 20 tasks, but I would rather see something comparing to the other methods from Table 1.
b) Thank you for being transparent in your experiment section about the parameter costs of your method. How does the additional parameter costs compare to other methods?
c) How does computation costs compare (both training time costs and inference time costs) to other methods?

**Limitations:**

Reasonable discussion on limitations is included.

---

> ### Author Rebuttal · Authors · 2023-08-10
>
> We would like to thank the reviewer for their time and feedback. We address the points individually and kindly ask the reviewer to increase their score based on our response.
>
> > “There seems to be large number of additional trainable parameters. While for 10 tasks, this is only 10/84 of the model size, it could double the necessary parameters for a longer task sequence.”
>
> **Response:** The mentioned 10/84 relates to the $LM$ untrained RP weights. This does not change as the number of tasks or classes grows. For DIL (Table 3), there is no parameter growth with tasks. For CIL, the number of parameters in the output head will double for any method (not just ours), if the number of classes doubles. Our weights per class is larger due to expansion from $L$ to $M$, but when using RP with $M=10000$, we would need ~4200 classes to reach half the ViT-B/16 size. Smaller $M$ can suffice, however, as indicated in Table SM5.
>
>
>
> > **Question a):** “How does the method perform on longer task sequences? For example, 20-tasks of ImageNet-R? There is one table in the SM with 20 tasks, but I would rather see something comparing to the other methods from Table 1”
>
> **Response:** We agree a comparison of $T=20$ for more methods will be valuable. The following table compares our approach to results extracted from ref [51] and will be added into Table SM4:
>
> | Method     | CIFAR100 | IN-R  | IN-A  | CUB   | OB    |
> | ---------- | -------- | ----- | ----- | ----- | ----- |
> | Ours       | **90.8**     | **75.4**  | **58.9**  | **89.7**  | **79.4**  |
> | L2P        | 70.96    | 56.25 | 40.71 | 58.23 | 60.19 |
> | DualPrompt | 72.04    | 69.25 | 40.95 | 67.46 | 64.39 |
> | ADaM       | 89.67    | 70.47 | 51.48 | 86.7  | 73.53 |
>
>
>
> > **Question b):** “How does the additional parameter costs compare to other methods?”
>
> **Response:** In comparison methods, L2P, DualPrompt and ADaM each use ~0.3M-0.5M parameters, while CodaPrompt uses ~4M parameters. SLCA trains all 84M ViT parameters. In comparison, for $K=200$ classes and $M=10000$ we use between 2M and 2.5M trainable parameters (depending on the PETL method), and 10M untrained parameters. We need to highlight that the random projections are not trainable. Therefore, the training overhead compared to other approaches is the difference in the size of the projection space from $L$ to $M$. We believe this increase in dimensionality is a trade-off for simplicity of implementation and low-cost training (e.g. the extracted features can be cached).
>
> > **Question c):** “How does computation costs compare (both training time costs and inference time costs) to other methods?"
>
> **Response:**  Inference speed is negligibly different to the speed of the original pretrained network, because both the RP layer and the final linear head are implemented as simple fully-connected layers on top of the underlying network. For training, Phase 1 trains PETL parameters using SGD for 20 epochs, on ($1/T$)'th of the training set, so is much faster than joint training. Phase 2 is generally only slightly slower than running all training data through the network in inference mode, because the backbone is frozen. The slowest part is the inversions of the Gram matrix, during selection of $\lambda$, but even for $M=10000$ this is in the order of 1 minute per task on a CPU, which can be easily sped up. In general, we believe the efficiency and simplicity of our approach compared to the alternatives is very strong. We will add remarks on this to the final version.

---

> > ### Comment · Reviewer_2MfY · 2023-08-15
> > **I am satisfied with the rebuttal response**
> >
> > I thank the reviewers for their rebuttal response and raise my rating to accept. Thank you for the hard work.

---

### Official Review · Reviewer_cPUm · 2023-07-24

**Soundness:** 3 good
**Presentation:** 2 fair
**Contribution:** 2 fair
**Rating:** 4
**Confidence:** 4

**Summary:**

The paper investigates the issue of continual learning using frozen pretrained vision transformers. The authors conduct a thorough analysis of potential limitations and strengths of continual learning methods that utilize pretrained models, supported by theoretical studies and derivations. Additionally, they introduce a novel algorithm called RANPAC, which incorporates a random projection layer with a nonlinear activation function, combined with a Parameter-Efficient Transfer Learning method. To assess the efficacy of their proposed solution, the authors conduct extensive experiments on various datasets such as Cifar and ImageNet, and across different scenarios including Class-Incremental Learning and Domain Incremental Learning.

**Strengths:**

The authors have successfully addressed the problem at hand and conducted a comprehensive set of experiments to assess the effectiveness of their proposed solution. The motivation behind their research is evident, and the current trend of expanding foundation models to the continual learning (CL) field makes their work particularly relevant.

Both the introduction and related works sections are clear and well-organized, providing a solid foundation for the study. The method description is well-supported by theoretical derivations, which are also included in the supplementary material for further clarity.

The presence of an ablation study to support the obtained results is highly commendable, as it enhances the reliability of their findings. Additionally, the authors' submission of code for transparency and reproducibility is greatly appreciated, further validating the credibility of their research.

**Weaknesses:**

The paper suffers from issues with clarity and organization. The current structure makes it challenging to follow as the authors have mixed background information with the method, resulting in an unclear narration. To improve the paper's flow, the authors should consider moving the "Overview and Intuition" subsection to the beginning of the method section. This would provide readers with a better understanding of the approach before delving into the technical details.

While the topic covered is of interest to the CL community, the contribution of the paper appears to be limited. Although the proposed algorithm is supported by derivations and theoretical foundations, it seems like a combination of existing tricks in CL to mitigate the shift towards new classes. As a result, the novelty of the approach is limited.

Regarding Equation 1, the usage of the Gram matrix updated over time, while effective, does not appear to be a strong innovation compared to techniques like LDA.

Table 1 raises concerns about the upper bound, particularly the performances of the joint linear probe. The results are surprising and not entirely convincing.

The message of figure 2, is not clear and easy to get. The left part of the figure is not clarified in the text.

**Questions:**

- The lack of an analysis of different PETL methods is a notable gap in the paper. Since Algorithm 1 is designed to work with any PETL method, it would have been interesting to observe and compare the performances of different PETL solutions.

- In section 3.2, the authors discuss the impact of RP, M. However, they fail to demonstrate the impact on performances adequately. It would have been beneficial to compare models with the same phi but varying M from 1 to a significantly larger dimensionality (ideally infinite from the claim of the authors).

- The authors claim that M=2000 is a good choice in section 3.2, but during experimentation, they raised it to 10000. The reason for this discrepancy should be addressed to provide clarity on the choice of M and its impact on the results.

- To further assess the effectiveness of the proposed solution, it would be valuable to study larger first task scenarios, such as 50-10, or more challenging scenarios like 50-5/50-2.

**Limitations:**

The authors analyzed the limitations of their method in few lines. before concluding the paper. This section should be definitely expanded.

---

> ### Author Rebuttal · Authors · 2023-08-10
>
> We would like to thank the reviewer for their time and feedback. We address the points individually and kindly ask the reviewer to increase their score based on our response.
>
> > “...the authors have mixed background information with the method ... should consider moving the "Overview and Intuition" subsection to the beginning of the method section.”
>
> **Response:**  We agree with the recommendation and in our final version will introduce an explicit separate Background section, and move material into it.
>
> > “... it seems like a combination of existing tricks in CL to mitigate the shift towards new classes. As a result, the novelty of the approach is limited.”
>
> **Response:** We respectfully disagree. The novelty and primary contribution is the introduction to pretrained models of frozen untrained RP (random projection layer to dimension $M\gg L$) with nonlinear activation *which has never been done in the continual learning context*. This is *not* an existing trick in CL with deep neural networks.  For the CL community, frozen RP weights is a fresh strategy that may inspire its use within other CL methodologies, because forgetting cannot happen in those weights.
>
> > “…the usage of the Gram matrix updated over time, while effective, does not appear to be a strong innovation...”
>
> **Response:** We agree, which is why Pages 1-2 (lines 91--103) do not list this as a methodological contribution. Three reasons for using the Gram matrix are explained in the Appendix, Sec.B.4. The choice  helps our conceptual contributions illustrated in Fig.2. Moreover, it enables the links to theory in Sec B.3 and B.4.4, that lead to our interpretation of the method as decorrelating classprototypes (which is absent in NCM) and our use of ridge regression (important for our empirical results).
>
>
> > Surprising upper bound results in Table 1.
>
> **Response:**  The "upper bound" for continual learning performance is *joint fine-tuning*. Our results for joint fine tuning are in Table 2. In Table 1, we report only joint linear probe results (with frozen backbone), and these are not an "upper bound." We will clarify this when we mention upper bound on line 97, and within the Table captions.
>
>
> >  “The message of figure 2, is not clear and easy to get. The left part of the figure is not clarified in the text.”
>
> **Response:** Fig.2 shows that our method reduces correlations between the class-prototypes of different classes (right side), thus creating better class separability (left side). We will add this to the caption. The left side of Fig.2 is explained on Page 4, lines 175-182, which is the next paragraph after the right side was explained. We will join these paras together.
>
> > **Question 1:** “...compare the performances of different PETL solutions..."
>
> **Response:** Analysis of the 3 PETL methods is provided for 7 datasets in the Appendix, Sec.F.6 (lines 766-771 for analysis) and Fig.SM7 shows comprehensive comparisions. We followed ref [51] for PETL. Although our overall methods are different to [51], like them, we found that the best performing PETL method is very dependent on the dataset (and pretrained weights). E.g. we found for CIFAR100 and $T=10$ that AdaptMLP gives better results than SSF or VPT, for Cars VPT is best, and for ImageNet-A, SSF is best (lines 766-771).
>
> > **Question 2:** “It would have been beneficial to compare models with the same phi but varying M from 1 to a significantly larger dimensionality....”
>
> **Response:** We explore the scaling with $M$ in the Appendix, Sec.F.5. Table 5 shows results for split CIFAR100 for $M$ increasing from $100$ to $15000$ for the same $\phi$. The table shows diminishing returns as M goes past ~$5000$. Also, Fig 3 (left) shows data for $M=5000$, $10000$, and $15000$ with the same $\phi$.
>
> > **Question 3:** “The authors claim that M=2000 is a good choice in section 3.2, but during experimentation, they raised it to 10000. The reason for this discrepancy should be addressed to provide clarity on the choice of M and its impact on the results”
>
> **Response:** In Sec.3.2, we arbitrarily chose $M=2000$ solely for the purpose of illustration in Fig.2. We made no claim about this choice being “good”. For consistency, we will update Fig.2 to use $M=10000$, but there are no perceptible differences in this visualization.
>
> > **Question 4:** “...it would be valuable to study larger first task scenarios, such as 50-10, or more challenging scenarios like 50-5/50-2.”
>
> **Response:** In the Appendix, Sec.F.3, we compare $T=5,~T=10$ and $T=20$. The $T=5$ scenario produces a larger first task. Table 4 shows that performance is better when the first task is larger. The reason is that Phase 1 adapts the pretrained model to a larger chunk of the overall dataset. Also note the column “No PETL” in Table 4 that stands alone without a choice of $T$. This is because Phase 2 has no dependency on the size of task increments. We mention this on lines 242--243, and with further details in Sec.F.3, i.e. in Phase 2, final accuracy is invariant to the order in which a completed set of data from all $T$ tasks is used. This property follows from Eqn (3), the use of frozen untrained RP weights, and the use of class-prototypes. This combination enables one class at a time to be added, with prediction outputs for that class unaffected by the subsequent addition of more classes.
>
> > “The authors analyzed the limitations of their method in few lines. before concluding the paper. This section should be definitely expanded.”
>
> **Author Response:** Thanks for the comments. We will expand the discussion of limitations in the camera-ready version.

---

> > ### Comment · Reviewer_cPUm · 2023-08-16
> > **Reviewer Response**
> >
> > Thanks to the authors for the detailed answer.
> > However, I still have some doubt about novelty and the effectiveness of the method.
> > - Even if the RP with M>>L is claimed as a huge contribution, I have still some concerns about the intuition  and if this is sufficient as it is.
> > - In SM section F3, the dimension and the number of classes in the first task is not specified.
> > - The motivation behind section SM F4 is not clear, why do the authors present the result for Task Agnostic Continual Learning? Task Agnostic Continual Learning and Class Incremental Learning should be the same from my point of view.

---

> > > ### Author Response · Authors · 2023-08-19
> > >
> > > Thankyou for your time and the additional remarks and suggestions. If you find our clarifications satisfactory, we kindly ask to consider increasing the score accordingly.
> > >
> > > > "... the effectiveness of the method."
> > >
> > > **Response:** The strong effectiveness of our method is evidenced by the second-last row of Tables 1 and 3. Our results show that introducing RP leads to at least 10\% reduced error rate in 9 out of 10 datasets, e.g. CUB, Core50 and DomainNet has 26%, 32% and 27% improvement respectively.
> > >
> > > > "Even if the RP with M>>L is claimed as a huge contribution, I have still some concerns about the intuition and if this is sufficient as it is."
> > >
> > > **Response:** Our underlying intuition is discussed in Section 3.3 (especially Lines 258-271, page 6, and Figure 3).  To summarize, we observed that Class Prototypes (CP) formed from the pretrained model's embeddings can be made more linearly separable if interactions between features are utilized. Since training a layer of weights where we insert RP is not feasible for CL (as it results in catastrophic forgetting), we instead propose RP followed by nonlinear-activation in order to create a transformed set of features that is more linearly separable using a CP method than features directly extracted from a pretrained model, as confirmed in Figs 2 and 3. As an additional contribution, we also show, for the first time, why CP strategies for CL benefit from decorrelation using second-order statistics, i.e. compared to simple NCM, similarities between CPs and comparison embeddings become better calibrated, resulting in enhanced linearly separability. We also show that our approach has high generality, with strong enhanced performance on 10 datasets, three scenarios (CIL, DIL and task-free), and for both transformers and CNNs  (see Results).
> > >
> > >
> > > > "In SM section F3, the dimension and the number of classes in the first task is not specified.
> > > "
> > >
> > > **Response:** In our rebuttal when we described having a larger first task with $T=5$, we meant this relative to the $T=10$ case, not that the first task had more classes than the other tasks. In all CIL experiments, all tasks (including the first) have identical sizes for the same $T$. E.g. if $K=200$ and $T=10$ then all $10$ tasks have 20 classes. But if $T=5$ then all tasks have 40 classes. So our point was that with Phase 1 adaptation, having 40 classes in the first task for $T=5$ explains the stronger results overall than for $T>5$.
> > >
> > > > "The motivation behind section SM F4 is not clear, why do the authors present the result for Task Agnostic Continual Learning? Task Agnostic Continual Learning and Class Incremental Learning should be the same from my point of view.
> > > "
> > >
> > > **Response:** We use an identical "task agnostic" scenario to that introduced in ref [56]. Others in the literature have used the term "task free" instead (Shanahan et al, "Encoders and Ensembles for Task-Free Continual Learning", 2021) and we will clarify this in the camera ready. In summary,  in the scenario of [56] and our Section F4, the set of classes available during *training* changes randomly with no way to define task boundaries. In contast, for CIL, although *inference* is task agnostic, *training* is applied to disjoint sets of classes, described as tasks.

---

### Decision · Program_Chairs · 2023-09-21

**Decision:**

Accept (poster)

**Comment:**

After discussion and rebuttal, the paper had mainly positive scores (4,5,7,7). The AC agrees with the positive reviewers and recommends acceptance. The authors should include the answers in preparing for the final version, especially the longer task sequence results and improved discussion on #Params and computational costs.